# Dynamics and Processes on Laser-Irradiated Surfaces

**DOI:** 10.3390/nano13030379

**Published:** 2023-01-17

**Authors:** Juergen Reif

**Affiliations:** Brandenburgische Technische Universität—BTU Cottbus-Senftenberg, Platz der Deutschen Einheit 1, 03046 Cottbus, Germany; reif@b-tu.de

**Keywords:** laser-matter interaction, laser ablation, incubation, LIPSS

## Abstract

The modification of solid surfaces via the impacts of intense laser pulses and the dynamics of the relevant processes are reviewed. We start with rather weak interactions on dielectric materials, based on non-linear absorption across the bandgap and resulting in low-level local effects like electron and individual ion emission. The role of such locally induced defects in the cumulative effect of incubation, i.e., the increase in efficiency with the increasing number of laser pulses, is addressed. At higher excitation density levels, due to easier laser–material coupling and higher laser fluence, the energy dissipation is considerable, leading to lattice destabilization, surface relaxation, ablation, and surface modification (e.g., laser-induced periodic surface structures). Finally, a short list of possible applications, namely in the field of wettability, is presented.

## 1. Introduction

Very soon after the first realization of lasers, more than 50 years ago [1], these new sources of very intense, strongly directed energy delivery found great interest in view of their applications. A particular focus turned to materials processing [2,3,4,5], starting with the utilization of the transformation from light to thermal energy. Here, the essential advantage is the strong localization of the energy input to the irradiated area and the relatively small surrounding region of heat diffusion (heat-affected zone), together with—for pulsed lasers—comparably rapid processes. The main effects were local material melting (and rapid resolidification) and evaporation, resulting in phase transformation (e.g., laser hardening) and material removal (e.g., laser cutting, laser ablation).

After the thermal action, the next impact considered was due to the high electric fields in the irradiated area, and due to very high intensities at shorter wavelengths (around 1 µm and shorter) in particular. These high fields result in the modification of the materials’ electronic system, both transiently (e.g., harmonic generation [6]) as well as permanently, i.e., via ionization (resulting in plasma formation [7]) or bond breaking (photochemical laser ablation [8]). It was shown that the plasma formation on metallic targets irradiated by 10 ns laser pulses consists of a combination of surface evaporation and dielectric breakdown in the vapor during the same pulse [9].

Taking the complexity of the processes observed on laser irradiated materials, it appears reasonable to consider the dynamics of such interactions in more detail. Most studies in this context have been performed with ultra-short laser pulses with durations of several femtoseconds, a time scale comparable to the relevant time scales in solid materials [10].

## 2. Transient Modification of the Electronic System

Already at laser intensities of 10^6^ W/cm^2^, which are typical for non-linear optics at the nanosecond time scale, the electrical field strength amounts to 10^7^ V/m, which are non-negligible in comparison with inner-atomic fields at the order of 10^11^ V/m (H-atom: E_binding_ = 13.6 V/Bohr’s radius), corresponding to a laser intensity of 10^16^ W/cm^2^. Consequently, the electronic system of the material will be heavily perturbed transiently, usually due to strong, transient polarization. To the best of our knowledge, this polarization is only present during the irradiation by the laser pulses [11,12,13,14,15,16,17], which gives rise to a multitude of transient non-linear optical phenomena that are beyond the scope of this article and will not be discussed in more detail.

The high excitation of the electronic system also results in several longer lasting or even permanent modifications of the irradiated material:(1)Electrons can gain sufficient energy to escape from the material via ionization and (surface) charging;(2)The atomic binding can be softened;(3)The electron energy can be transferred to the phonon bath.

All of these effects of energy transfer have their intrinsic time scales, which will be discussed in detail below.

Once the energy is dumped into the surface and bulk of the material, further evolution is observed in the form of plasma formation, local heating, ablation, phase explosion, or morphology modification. Further, the material realignment can back-result in a permanent adaptation of the electronic system, and accordingly in light–material coupling (incubation).

An important tool used to study the dynamics of all processes is the pump probe experiment [18], whereby the system is “prepared” by a first laser pulse and then “interrogated” by secondary, delayed pulses with variable delays. With ultra-short laser pulses, a time resolution range from (few) femtoseconds to nanoseconds is achieved using optical delays. For longer time scales, typically, electronic delays are applied.

## 3. Permanent Material Modification

In this chapter, we review the different effects and their dynamics in more detail. However, the modification of the surface morphology will be fully covered in its own dedicated chapter.

### 3.1. Subthreshold Interaction

We will start with concentrating on a laser irradiation that is well below the typical threshold for massive material removal, called “laser ablation” (e.g., via phase explosion or plasma ignition and erosion [10,19]). However, we will not consider photochemical bond-breaking in organic, polymeric materials observed for UV laser pulses [20], which is a complex topic of its own and is outside the scope of this review.

#### 3.1.1. Instantaneous Effects: Ionization and (Surface) Charging

First, we consider—mostly for wide-bandgap dielectric targets—very moderate interactions, whereby the individual target electrons are excited across the bandgap via multi-photon absorption or electron tunneling [21]. Consequently, the electron density in the conduction band is still weak. Nevertheless, sporadic electrons and consequently ions are emitted from the surface, since for dielectrics the lower edge of the conduction band is close to the vacuum level. We prefer to call this low-level particle emission, whereby only charged particles are removed from individual surface sites, “desorption” instead of “ablation” occurring with the formation of distinct ablation craters.

Immediately upon the arrival of the laser pulse, there is an instantaneous escape of (surface) electrons [22]. The surface becomes positively charged because the electron mobility is not sufficiently high to immediately compensate for the removed negative charge. This results in dielectric instability and the surface relaxes via a Coulomb explosion [23,24,25,26,27,28,29], releasing positive ions—and sometimes even larger molecular clusters [30] (Figure 1)—which can reach high (monochromatic) kinetic energy levels corresponding to tens of km/s [25,31] (Figure 2).

It should be mentioned here that similar results have been obtained from silicon [32], although the band structure is different from the insulators and the Coulomb explosion seems not to explain the observations (despite previous reports of Coulomb explosions from Si [33]). However, it has been claimed that the ion emissions occur from a state of thermal disequilibrium.

Depending on the degree of ionization and the amount of desorbed positive ions, a space charge can build up, translating into a saturated desorption rate and slowing down the ions. This is characterized by a bimodal fast–slow ion distribution [31,34] (Figure 3).

It remarkable that from the dielectrics results we also observed the emission of much slower *negative* ions [29,31,35] (Figure 4) which are much slower.

Whereas the emitted particles are mostly charged ions, the emission of neutral particles is much less abundant, indicating that thermal emission (evaporation) does not play an important role at the studied low excitation densities below the macroscopic ablation threshold. 

The emission of a large number of particles with substantial kinetic energy levels is associated with considerable recoil pressure onto the sample. In silicon, this local pressure load in the GPa range results in phase transformations in the crystal lattice [36]. The resulting new phases (e.g., hexagonal, bcc, or rhombohedral silicon) can be detected via micro-Raman spectroscopy [31,32] (Figure 5).

#### 3.1.2. Local Defects and Incubation

Even though the interaction considered so far is very weak and below the ablation threshold, the desorption of surface constituents results in a modification of the residual surface by introducing local defects such as color centers [37] or self-trapped excitons (STEs [38]). These will induce local states within the bandgap, which can locally increase the absorption probability by reducing the multi-photon order. In fact, when applying repetitive multiple pulses, the well-known phenomenon of incubation is observed, i.e., a decrease in the desorption or ablation threshold with the increasing number of incident pulses [39]. Most often, the effect is evaluated using a purely statistical approach [40], connecting the *N*-pulse ablation threshold *F_N_* to the single-pulse threshold *F_1_* via the relation *F_N_* = *F_1_ N ^S^*^−1^, where *S* < 1 is a fit parameter.

Instead, we proposed a different approach to evaluate the incubation effects based on the idea presented above that repetitive laser impact results in an avalanche-like increase in local defects and a corresponding decrease in the threshold fluence [41,42,43]. This is accounted for by an exponential decay *I_N_* = *I*_1_ exp(-α*N*) (here, we refer to the laser intensity, *I*, and not to the fluence, *F*, since the electric field in the irradiated zone is important for multi-photon or tunneling ionization). In Figure 6, the effects of incubation and the fit by our exponential model are presented.

### 3.2. Near-Threshold Interaction

In this part, we will deal with slightly higher excitation of the target, so that not only does instantaneous desorption to be considered, which in principle relaxes the excitation immediately. Instead, the dissipation of the excitation and its relaxation can become important. Additionally, the interaction of the residual laser pulse with material already excited during the first part of the pulse (plasma formation [7,44], hot electron excitation via inverse bremsstrahlung [45], etc.) may play an important role, resulting in additional energy being input into the target. 

Laser-induced formation and the heating of expanding plasma are very complex topics in themselves; therefore, they are beyond the scope of this review. Instead, we will concentrate on effects mostly confined to the target surface region. One particular focus is the transfer of electron energy to the target lattice and its consequences.

#### 3.2.1. Hot Electron Excitation and Energy Dissipation

Differing from free particles or the surface escape depth [31], excited electrons are free carriers in the conduction band, where they can absorb additional energy (“free carrier absorption” [46]). They can generate further electrons via impact ionization [47], can generate electron–hole pairs through collisions with valence band electrons, and can increase the conduction electron density via Auger processes. Thus, very substantial amounts of energy can be absorbed in an avalanche process, and a high electron density in the conduction band can be created. In fact, it has been shown that dielectrics and semiconductors can exhibit metal-like optical properties [48]. 

The energy deposited in the electron gas must dissipate to the target lattice. This occurs typically (if not through an electron–hole plasma) via electron–phonon collisions, and is usually considered using Anisimov’s two-temperature model [49], where the excitation first establishes an equilibrium of hot electrons, with the lattice still being in equilibrium at a significantly lower (environmental) temperature. Through electron–phonon collisions, the lattice is heated to well above the environmental temperature, with a typical time for this energy coupling being in the order of a few picoseconds [50].

During this transfer, i.e., before the phonon bath is heated, the hot electron gas may significantly influence the electronic binding between the target atoms. Therefore, the atoms are no longer held at their lattice sites but may “soften” the rigidity of the lattice. This has been shown using both simulations [51,52] and experiments [53,54]. Additionally, ultrafast melting in semiconductors [55], with time constants of a few 100 femtoseconds, can be attributed to this type of modification of inter-atomic forces.

Additionally, the bandgap may be modified, resulting in a change in absorption probability based on the laser wavelength. This is shown in Figure 7, where the desorption dynamics are investigated in pump probe [18] experiments at an intensity slightly below the desorption threshold; the emission of electrons and ions not only occurs with zero pump probe delay for overlapping pulses but is also observed after a finite pump probe delay of several hundred femtoseconds [56,57,58] (Figure 7). 

This can be understood by assuming that the signal around zero delay is due to the coherent interaction of pump and probe pulses, whereby the target cannot distinguish between the two pulses. Each one alone is not sufficient for appreciable desorption from a virgin surface, but their combined irradiation is double that of a single pulse, which is enough. The width is determined by the phase coherence time *T*_2_ [59] of the electron excitation. The delayed desorption peak can, therefore, be attributed to the modified target after dissipation of electron energy to the lattice (*T*_1_) having a lower desorption threshold, meaning that a single pulse energy is sufficient for desorption. Similar behavior has been observed on metal targets [56,57,58]. 

#### 3.2.2. Rapid Heating and Melting

In the process of electron–phonon energy transfer, the electron ensemble is cooled and the lattice is heated until the complete system is in thermal equilibrium. 

With classical thermodynamic behavior, first the absorption volume is heated and then the heat is propagated further down into the volume via heat diffusion. If the energy is high enough, this means that a melt front is gradually penetrating the target. This process is called “heterogeneous melting”.

At high electron energy levels, however, hot electrons can ballistically intrude far into the target volume and can deposit their excitation equally there, with simultaneous subsurface boiling in a large target region, called “homogeneous boiling” [59]. Since they are surrounded by non-excited cold matter, the strongly confined boiling centers experience a significant pressure increase from their environment, resulting in an increased boiling temperature. Therefore, the excited volume is finally “super-heated” [60] within a very short time, with heating rates of up to 10^14^ K/s [10]. This confined excitation must also relax in a very short time in a “phase explosion” [61]. It should be noted here that although this process can be fully explained by thermodynamic principles, it is not really in thermodynamic equilibrium. Therefore, we may call these effects “hyperthermal” [62]. Hyperthermal processes are characterized by a very rapid build-up of a highly energetic volume surrounded by a cold lattice. This corresponds to a steep gradient in atomic order, resulting in very fast relaxation; it is too fast to proceed in thermal equilibrium, as it typically occurs in a time shorter than the equilibration time (usually involving a few generations of phonon–phonon collisions, i.e., several tens of picoseconds).

## 4. Modification of Surface Morphology

In this chapter, we will consider the formation of (regular) nanostructures at the target surface, which are typically termed “laser-induced periodic surface structures (LIPSS)”. Interestingly, similar surface modifications are also encountered upon surface irradiation with energetic ions [63]. Hence, the laser results will be also discussed in comparison to the ion beam results, assuming that in fact that the high and fast energy input is mostly responsible for the structural formation, independent of the type of energy source.

### 4.1. Experimental Structural Formation Results

#### 4.1.1. Laser Impact (LIPSS)

Laser-induced periodic surface structures have been intensely studied on metals, semiconductors, and dielectrics for more than six decades [64,65,66,67,68]. After several attempts to model the phenomenon in the 1980s—which will be discussed in more detail in Section 4.3—the topic somewhat lost public attention. In fact, in laser matter processing, the surface structures were considered to be more or less annoying and unwanted. 

Only about 20 years ago was the topic called up again [69,70], and from then on it has found ever increasing interest (cf. Figure 8) (for reviews, see [71,72]). 

In fact, many (quasi)periodic structures have been observed, ranging from (quasi)parallel ripples to larger structures such as grooves, cones, and islands [73] (cf. Figure 9). For the special case of ripples, the nomenclatures of “high spatial frequency LIPSS (HSFL)” for structures with a spacing well below and “low spatial frequency LIPSS (LSFL)” for structures at about the laser wavelength or slightly above have become widely accepted [72] (cf. Figure 9). An important parameter for such structural formation is the laser polarization [74]. This usually determines the orientation of the ripples (it must be noted, however, that this influence can be overridden by macroscopic surface damage). In general, e.g., HSFL and LSFL are aligned perpendicularly to each other (Figure 10) [69]. It should be noted, however, that the actual correlation, i.e., whether the HSFL or LSFL are perpendicular to the polarization, seems to depend strongly on the material [72]. Obviously, other than linear polarization, circular or elliptical polarization gives rise to different types of structures such as dot patterns (circular) or ripplets, which are limited by bifurcations (elliptical) [75]. Their length is proportional to the ellipse’s eccentricity, and they are aligned according to the long polarization axis. This is shown in Figure 11 [62].

In Figure 12, a special example of LIPSS formation on CaF_2_ is shown [31,70]. Note that it is taken from *one single* interaction spot. In that spot, HSFL and a parallel, coarser structure can be observed together, which do not correspond to typical LSFL. The main control parameter appears to be the local irradiation dose, with an abrupt transition between both structures. This induces the question about the influence of fluence or even accumulated energy [77]. For a pulse repetition rate of 1 kHz, the energy of the subsequent pulses can be considered to accumulate before any considerable relaxation [78,79] (cf. Section 4.2). Therefore, in Figure 13, the accumulated irradiation dose (number of pulses × pulse energy) is considered to illustrate this effect.

##### Texture of Near-Surface Volume

It is interesting to question how deep the material modification extends to the immediate subsurface region of the target, particularly in view of the formation mechanisms (cf. Section 4.2. below), i.e., whether the structural formation is purely ablative or involves the material more actively. With high-resolution transmission electron microscopy (HRTEM), we studied cross-sections of Si(100) modified via multi-shot irradiation with ≈100 fs @ 800 nm laser pulses at intensities below the single shot ablation threshold [80]. Earlier experiments have shown that the irradiation of semiconductors by ultrafast laser pulses can result in both amorphization of the crystalline material [81] and crystallization from the amorphous state [82]. For the HRTEM investigations, the LIPSS spot (Figure 14a) was covered using a Pt protecting layer before two thin lamellae were cut out (inserts), one at the spot’s center with strong surface modulation (Figure 14b), and one at the spot’s edge (Figure 14c) with a part *outside* the visibly modified area. As can be seen in Figure 14b, the ripple crest is (multi)crystalline but not commensurate with the target bulk material. In the valleys between the ripples, we found amorphous silicon. Interestingly, outside the visibly modified spot (cf. SEM), there is a quasi-continuation of the surface modulation, consisting of shallow dimples in the crystalline surface filled up with the amorphous material. In summary, the rippled area consists of both amorphous and crystalline—although not homomorphous—materials.

#### 4.1.2. Ion Beam Impact

As indicated before, very similar structures were obtained via ion beam irradiation [63]. Figure 15 gives an impression of the typical results [83]. Figure 16 and Figure 17 show direct comparisons between laser [69,84] and ion-beam-induced surface structures [82,85].

### 4.2. Dynamics of LIPSS Formation (Feedback)

In this section, we approach the dynamics of LIPSS formation in three ways: first, we consider the effects of separated pulse pairs (very similar to the pump probe experiments but with a “postmortem” analysis; second, as an extension of the first approach, the effect of the pulse repetition rate is considered; third, the pulse-to pulse evolution of a single spot is studied.

#### 4.2.1. Double- or Multiple-Pulse Exposure

There have been double-pulse investigations by many groups in recent years (e.g., [72,86,87,88,89,90,91]. A typical experimental arrangement is shown in Figure 18 [86]. One of the main questions is whether the first pulse of the pair is essentially responsible for the structural formation (and in turn the orientation) or whether it only prepares the target to be more susceptible to the second pulse, which had the greater influence. This may help for the future discussion of models in Section 4.3. Therefore, it should be noted that there are still some discrepancies between the results found by different groups that have to be resolved [72,92,93]. Interestingly, for simultaneous double pulses (non-collinear, zero time delay, parallel polarized), no signature of interference-induced transient dynamic index grating [12] could be observed in the surface modification.

A specific case of multiple-pulse exposure (which is typically necessary for surface modifications with subthreshold fluence [19,72,94,95,96,97], resulting in positive feedback) can help in understanding the structural formation dynamics, namely the variation of the pulse-to-pulse temporal separation by varying the repetition rate [78]. Therefore, it could be shown that the feedback from the preceding pulse on both the modified area and the ripple density or spacing is significant, even for a pulse separation time of 1 s (Figure 19), with the modified area and ripple spacing decreasing with increasing pulse separation. Numerical simulations show that this effect should be due to pulse-to-pulse heat accumulation and very slow cooling rather than the hot carrier dynamics, which should relax much faster [78].

As indicated before (Section 3.2.1, Figure 7), for double pulses the energy input during the first pulse will change the target’s susceptibility to absorption during the pulse separation time. Interestingly, the LIPSS modulation is not simply ablative but also extends above the initial surface level, as shown in Figure 20, where conventionally cleaned silicon (100) wafers were irradiated in an ultra-high vacuum (10^−9^ mbar) by 50 pulse pairs, with each pulse around the ablation threshold [91]. An investigation of the LIPSS spots using µ-Raman spectroscopy (Figure 21) revealed significant material changes for both the zero delay (coinciding pulses, total fluence of 4.7 TW/cm^2^) and 10 ps delay (separated pulses, 2 × 2.3 TW/cm^2^). All spectra are dominated by the TO–phonon peak of crystalline silicon with a Raman shift of 520.7 cm^−1^. In contrast to the result shown in Figure 5 (120,000 pulses @ 0.64 GW/cm^2^), no distinct polymorphs can be observed; only a new peak at a Raman shift of 485 cm^−1^ shows up, which was previously attributed to the presence of nanoparticles [98], with increasing intensity towards the spot’s edge, especially with zero delay or high fluence, which is weaker in the 10 ps delay area. 

On the other hand, the TO main peak is red-shifted towards the spot’s edge, indicating increasing tensile stress. This is significantly more pronounced for the separated pulse pair. Further, in this case we found an increased contribution of the amorphous material towards the spot’s edge, as indicated by the broad shoulder of the main peak.

A closer inspection of the main peak shifts (Figure 22) indicates that the spot and its vicinity experience tensile stress up to about 1 GPa, which is particularly important for separated pulse pairs.

#### 4.2.2. Pattern Evolution

As indicated before (Section 4.1.1, Figure 9, Figure 12 and Figure 13), the irradiation dose (pulse energy × number of pulses), i.e., the incident or absorbed energy, obviously plays an important role in the particular feature size of the generated structures. This is particularly obvious in Figure 13, where the dose is changed by the increasing number of pulses at a fixed fluence level. Here, certainly the effect of the feedback is important. Similar to general incubation, the generated structures may both increase the susceptibility and amplify or further evolve the modified surface structure.

To shed light on this influence, different experiments have to be considered: (1) different spots with different exposure conditions have to be analyzed post-mortem using different techniques, which then need to be compared; (2) a single spot has to be irradiated by an initial dose, analyzed in situ, then further irradiated (at identical geometrical conditions), analyzed again, and so on.

##### Irradiation Dose Dependence

Due to the spatial beam profile (typically Gaussian), even in a single laser spot (after an arbitrary number of incident pulses) a corresponding variation in the local dose can be observed over the cross-section (Figure 12, Figure 23 and Figure 24). This can be considered also as *intra-pulse* feedback [72]. Particularly interesting is the very regular variation shown in Figure 24; the structure changes in distinct steps, with the feature size approximately doubling at each step [99] (period doubling).

On the other side, the dose is also increased more generally by a series of repetitive pulses. In fact, here the role of the *inter-pulse* feedback [72] is more evident, which has been widely studied [72,79,94,95,96,97]. As expected, an increasing pulse number (and irradiation dose) results in an increased modified area size. At the same time, however, in most cases the LIPSS feature size increases, as shown in Figure 13, Figure 25 and Figure 26 [77,78,100].

The regular increase in the modified area can be considered as reflecting a reduction in threshold energy similar to the effects of incubation [42]; if the Gaussian spatial profile at the first pulses only exceeds the modification threshold at the very center, an exponential threshold reduction involves larger parts of the beam cross-section. 

This is indicated by Equation (1) and Figure 27 [101]:*A* = *A_max_* (1 − exp {−k·θ}) = *A_max_* (1 − exp {−k′·N})(1)
where *A* is the modified area, *A*_max_ the maximum beam cross-section (at the “bottom” of the beam profile), *N* the number of incident pulses, Θ = *N* × *E_pulse_* is the total dose with single-pulse energy *E_pulsee_*, and *k*, *k*′ are coupling constants.

##### Genesis of Surface Structures

To follow the origin of laser-induced morphology modifications, very low doses (very low fluence or few pulses) have to be studied [77,102,103]. The experimental results for CaF_2_ irradiated at 25% of the ablation threshold (8.2 TW/cm^2^, 800 nm, 100 fs) show that at a very low dose (10 pulses), no *regular* structures developed [102]. Only the electron reflectivity in the irradiated spot was increased so much, that it can be distinguished in the SEM image in Figure 28a. 

However, as a first signature of the morphological modification, a large number of (semi-spherical) bumps occurred, which were swollen above the pristine surface level, with diameters of a few µm (details in Figure 28b,c) and distributed over a large area, even outside the visible spot. Even though this seems to indicate the redeposition of ablated debris, the total volume of all bumps appears to be much larger than for all ablated material. This supports the idea of material swelling [104]. Additionally, it should be noted that the “visible spot”, i.e., the region of increased electron reflectivity may well be smaller than the area that is actually irradiated (at even lower doses in the spot wings). At a 100-fold higher dose, the first indication of more important and regular morphology changes was observed (Figure 29).

Experiments on silicon samples yield more insights into the dynamics of laser-induced structural formation. In a first set of experiments, the evolution of a single spot during the irradiation was studied with a fluence at about 50% of the ablation threshold. For this purpose, the target was kept in a fixed position with respect to the laser beam [103]. After several laser pulses, an AFM was slid over the sample for the in situ analysis, then removed to apply additional pulses to provide a well-defined series of laser pulses (Figure 30). A fixed marker on the sample, close to the irradiated spot, later allowed for the individual AFM micrographs to be stitched together (Figure 31). Clearly, the same structure and spot area evolution can be observed as for the multiple-spot mode. In particular, a *continuous* evolution becomes evident.

Irradiation at the ablation threshold (multi-spot [77,103]) yields similar results; for a single pulse, a periodic surface morphology modification can already be observed [103] (Figure 32), extending above and below the pristine surface. The wavelength of ≈625 nm is close to the laser wavelength. The structure is, however, *not* imposed by the laser polarization but instead constitutes a system of concentric rings determined by the spot circumference. Note that depending on the local fluence, less swelling of the ridges occurs in the spot’s center, resulting in a mean depression of that area by a few nm (the darker area in the AFM picture). The modulation depth or height is stronger at lower fluence levels in the spot’s slope (note the dark deep holes in that slope region, with an aspect ratio of about 0.1).

The influence of increasing pulses in multi-spot experiments (Figure 33) [77] is similar to the single-spot result shown in Figure 31, with an evolution from very weak formation at a dose of 10 pulses (Figure 33a) over a combination of HFSL and LFSL at 50 pulses (Figure 33b), and a deep crater with LIPSS modulations after 1000 pulses (Figure 33c). As with the single-shot result (Figure 32), at low doses deep dark holes can be observed, marked by the white arrows in Figure 33a and shown in detail in Figure 34. Note that these holes are the nuclei of concentric ripples, whereas the general ripple orientation is polarization-determined.

A closer inspection of the ripples from Figure 33, shown in Figure 35, shows that with increasing doses, the initial nano- and micro-ripples (550–750 nm) perpendicular to the direction of polarization (Figure 35a) coalesce into perpendicular macro-ripples (1.25–3 µm) (Figure 35b), interlinked by ladders of remaining micro-ripples. At high dose (Figure 35c), nano- and micro-ripples can be observed only at the spot’s edge. The perpendicular coarser features break up into slightly coarser short ripplets, again perpendicular to the direction of polarization. In Figure 33c, the ripplets closer to the spot’s center (at still higher dose than at the edge) develop to pillars extending deep into the ablation crater.

### 4.3. Modeling of Surface Modifications

Based on the seminal work of van Driel, Sipe, and coworkers [66], as described in detail by Bonse and coworkers [72,105] and triggered by the apparent similarity of LIPSS to typical interference patterns, a widely promoted model relates the structural formation to modulated ablation as a consequence of an inhomogeneous laser fluence distribution on the surface. The underlying pattern is attributed to an interference between the incident wave and a secondary wave due to scattering of the surface roughness or laser-induced surface polaritons. This electromagnetic light localization has been basically simulated using finite-difference, time-domain (FDTD) simulations [106]. Generally, this transfer from the irradiation pattern to the target surface can be classified as being holographic (or lithographic).

However, this approach cannot account for many of the observations described: e.g., the periodic swelling *above* the pristine surface (cf. Figure 20, Figure 28, and Figure 32), the polarization-independent concentric LIPSS around holes or along the spot boundary, period doubling (Figure 12, Figure 24, and Figure 26), or bifurcations (Figure 36, [107]). Further, periodic structures could be observed after irradiation with femtosecond white light pulses [108], without any prevailing wavelength, which are necessary for interference effects. 

Even more important is the close affinity with the structures formed upon irradiation with directed ion beams (cf. Figure 15, Figure 16 and Figure 17), again without any defined modulation of the electromagnetic fields. The structures even resemble patterns found on the electrodes during arc discharge erosion [109]. These observations triggered the idea of *surface instability* upon interaction (Section 3.1.1). This idea is supported by investigations finding the formation of a transient soft surface state (Section 3.2.1 [51,52,53,54,55]), where the inter-atomic binding in the surface region is no longer well-defined. The surface instability stimulated an approach to adopt models for ion-beam-induced surface modification, relating the surface instability to the well-known hydrodynamic instabilities of thin liquid films and postulating self-organized structure formation during such instability. Figure 37 schematically compares the two approaches used for modeling surface structure formation [77].

#### 4.3.1. Outline of the Self-Organization Model

Initiated by Sigmund’s seminal theory of ion sputtering [109,110], Cuerno and Barabásí [63] considered the time evolution of the surface corrugation caused by a competition between surface roughening via particle desorption and smoothing due to atomic diffusion. Correspondingly [94], the time evolution of the thin-film modulation height, *h*, can be described by a rate equation of the KPZ (Kardar–Parisi–Zhang) type [111], containing an erosion term and a diffusion term:(2)$$\frac{\partial }{\partial t}h=-v\left(h\right)\sqrt{1+{\left(\nabla h\right)}^{2}}-K\Delta ²h$$
where *v*(*h*) is the surface erosion velocity, depending on the surface curvature, and *K* accounts for the thermal self-diffusion, which depends on the surface diffusivity, activation energy for self-diffusion, density of the diffusing atoms, and temperature. The solution of Equation (2) predicts the formation of periodic surface structures, with the orientation and wavelength, *Λ*, both depending on the incident energy deposition (proportional to the incident fluence, *F*, and diffusion coefficient, *K*). A gross estimate can then approximate *Λ* in Equation (3):(3)Λ=2π2KF

Figure 38 shows a scheme of the thin, corrugated liquid-like film considered in the model [112].

The non-local integro-differential in Equation (2) can be reduced to a partial equation of the Kuramoto–Sivashinsky type [113,114]:(4)$$\frac{\partial }{\partial t}h=-v_{0}+\nu_{x}\frac{\partial^{2}}{\partial x^{2}}h+\nu_{y}\frac{\partial^{2}}{\partial y^{2}}h-K\Delta^{2}h+\frac{\lambda_{x}}{2}{\left(\frac{\partial }{\partial x}h\right)}^{2}+\frac{\lambda_{y}}{2}{\left(\frac{\partial }{\partial x}h\right)}^{2}+higher\;orders$$

A simulation of the structure evolution [77] with the simulation time (i.e., duration of the instability) pursuant to Equation (4) (lower row of Figure 39) yields results that appear to be *qualitatively* very similar to the experimental dose-dependent results (upper row of Figure 39). A closer examination of Equation (4) reveals that it consists of a contribution of linear terms (with coefficients *ν*_x,y_) and non-linear terms (with coefficients *λ*_x,y_). This separation between linear and non-linear dependences is associated with a cross-over time, *t*_c_, separating the regimes, as indicated in the lower panels of Figure 39. Obviously, the simulation reproduces the evolution from long parallel ripples (around *t*_c_/2) to an array of coarser ripplets (around *t*_c_; cf. Figure 35) and a break-up into rough cones in the non-linear regime (*t* > *t*_c_).

Performing a *quantitative* comparison between simulation and experiment is, however, not straight forward. One reason is the role of the time evolution. In the model and simulation, it denotes the duration of existence of the instability (such as in ion beam experiments, where it is sustained by *continuous* feeding of the incident energy). In the laser experiments, the instability is not continuously evolving because the feeding energy is delivered by repetitive packets. Here, the accumulated irradiation dose determines the “duration” of the instability, considering the long persistence of inter-pulse feedback [100], as shown in Figure 19. Therefore, it appears justified to compare the increasing simulation times with increasing irradiation doses. The second problem for a quantitative comparison is the lack of detailed material data that are input for the simulation, e.g., the data for the thermal self-diffusion term, *K* (surface diffusivity, activation potential, etc.), the initial erosion velocity, *v*_0_, and parameters *ν*_x,y_ and *λ*_x,y_.

#### 4.3.2. Laser Polarization in the Self-Organization Model

Although the self-organization model can so far describe the physical phenomenon of regular structural formation from laser-induced surface instability, it is not yet able to account for the paramount influence of laser polarization. In the following sections, we extend the model correspondingly. To do so, the energy input must be considered in more detail. Following Sigmund’s theory of sputtering [109], the erosion velocity normal to the surface can be related to the energy input (Equation (5)):(5)$$v_{0}≅\xi \overset{}{\underset{V}{\int }}dr^{3}\psi \left(r\right)\varepsilon \left(r\right)\propto \xi F$$
where the integration is performed over the absorbing volume *V.* Here, *ξ* is a material parameter; *ψ* (*r*) is the fraction of input energy absorbed at *r,* including effects such as the losses from reflection, transmission, and effective absorption; *ε* (*r*) is the fraction of lattice energy (atomic kinetic energy) after electron–phonon relaxation (cf. Section 3.2.1.) at *r*, which reaches the ablation spot at the surface. The transfer probability, which is essential for the active surface energy *ε* (*r*) in Equation (5), is inversely proportional to the electrons’ mean free path, *l*, which can be estimated from the “universal curve” of the electron mean free path vs. the electron kinetic energy [115], as shown in Figure 40.

##### Anisotropic Energy Diffusion to the Surface

If there is a non-symmetrical distribution of electron kinetic energy around the interaction point, *r*, this results in an inhomogeneous collisional energy transfer. Consequently, assuming Gaussian velocity distributions for the electrons, the contributions of energy absorbed at *r* to the erosion velocity *v*_0_ can be approximated by Equation (6):(6)ε(r)∝l−2exp[(x′22σx′2)+(y′22σy′2)+(z′22σz′2)]

Here, *σ*_x′_, *σ*_y′_, and *σ*_z′_ account for the respective collision cross-sections along the directions in an excitation-related coordinate system (*x*′,*y*′,*z*′) (cf. Figure 41), which is inversely proportional to the corresponding free mean paths (cf. Figure 41b). Introducing this anisotropic energy distribution at the surface results in a corresponding anisotropy in the self-organized pattern [74,116,117,118].

##### Anisotropy Induced by Laser Polarization

One possible form of anisotropy is induced by the laser polarization. To evaluate *ε* (*r*), we have to consider the geometric situation at the target [74,117,118,119] (Figure 41). For this purpose, we introduce the laser-beam-related coordinate frame *x*′,*y*′,*z*′, in addition to the laboratory coordinate frame *x*,*y*,*z*.

Considering the fact that the electric laser field deforms the electronic binding potential, as is known from atomic physics [31], with a preference along the field polarization direction (Figure 42), the electrons’ kinetic energy is highest in the direction of polarization (analogously, despite the symmetric energy deposition by the Gaussian beam profile, there is an acceleration of conduction band electrons); this means the same is true for the shortest mean free path (cf. Figure 40) and highest collision cross-section and energy transfer values, explaining the paramount role of laser polarization in forming the structure’s shape and orientation. Simulations along these lines show excellent qualitative agreement with the corresponding experimental results (Figure 43) [74,116,117,118,119].

It should be noted here that not only the laser polarization introduces anisotropy. Other possible influences may be the coupling to surface plasmon–polaritons [120,121,122] and local defects at the surface (e.g., scratches [77]) or in the bulk.

Anisotropies can also be due to material properties, especially in multi-component thin films or layer systems such as those used in randomly filled computer hard disks [123,124], consisting of a glass substrate, a 65 nm non-magnetic metallic buffer base (containing Ti, Ru, and Al), a 30 nm magnetic layer (containing >50% Co), and a 5 nm polymeric cover layer [125,126]. The complex multi-layer system is schematically shown in Figure 44. For this type of target, it is expected that a considerable part of the incident energy will be confined within the magnetic multilayer stack. The energy transmitted to the metallic buffer layer (optically and thermally) should be rapidly dissipated due to the good thermal conductivity of the metal. Efficient etch stops in the interlayer boundaries [127] hamper purely erosive pattern formation of the holographic scenario.

In fact, already after the first pulse, the ablation spot reflects the layer structure more than the beam profile (Figure 45) [124]. Although the linear polarized incident beam profile is Gaussian (Figure 45b), there are two distinct levels of ablation across the irradiated spot: one outer ring where only the polymeric cover layer is completely removed, and a large, flat central area where part of the magnetic layer is removed (Figure 45a,d). The ring only represents the surface of the magnetic layer (i.e., the interface between the polymeric cover and the magnetic layer), obviously showing a nanostructured array of parallel lines of dots with a typical feature size of about 500 nm (Figure 45c).

There are two remarkable features about this spot:(1)The ablation spot does not reflect the (continuous) Gaussian beam profile. Instead, the ring is sharply bordered and only the polymeric cover is removed there, not any of the magnetic layer. Further, the ablation of the magnetic layer in the central disk is of about constant depth, disregarding the fluence variations across the beam profile. This suggests two distinctly different coupling or ablation thresholds;(2)The uncovered surface of the magnetic layer in the ring is regularly structured. The morphology looks very similar to the LIPSS. However, it is not compatible with any polarization influence.

The latter feature, i.e., the microstructures at the surface of the magnetic layer, can be attributed to the magnetic domains (bits) of the randomly filled disk (Figure 46). The transformation of the magnetic domains to a corresponding modulation of the morphology can be attributed to magnetostriction of the different magnetic domains [128,129]. In the central area of the spot (with the unstructured plateau), the fluence is just high enough to destroy the magnetic order (this can be considered a sufficient energy input to overcome the Co Curie temperature *T*_C_ = 1.394 K) but not to further ablate the film material.

The situation changes after multi-pulse irradiation when the surface instability is established. The magnetic order is fully spoiled, and self-organized structural formation becomes possible, as indicated in Figure 47 after ten pulses. Obviously, the instability comprises the entire magnetic layer thickness, whereas the non-magnetic buffer is only slightly affected.

## 5. Surface Functionalization

In recent years, the laser modification of the surface morphology has generated a substantial range of applications with respect to the wettability, optical properties, tribology, wear control, corrosion resistance, and templates for biological or technological thin films, sensors, and more [71,72,105,130,131,132]. Most important for such applications is the creation of larger processed areas instead of individual small spots at the µm scale. Additionally, the modification of electrical or chemical surface properties can affect the applications, e.g., the immobilization of adsorbed biomolecules.

### 5.1. Modification of Electric Surface Potential

Similar to the magnetic properties discussed above, the “soft state” of instability can also influence the electrical surface properties of the target; whereas in the magnetic case the Curie Temperature is exceeded, destroying the magnetic order, for silicon the dopant mobility is increased, resulting in modified dopant segregation [133] and a corresponding modification of the surface potential. This becomes evident following an investigation of the surfaces of the LIPSS of silicon using electrostatic force microscopy (EFM) and scanning Kelvin microscopy (SKM), respectively, as shown in Figure 48 [134,135]. The LIPSS formation impresses a corresponding nanostructured pattern on the target surface potential.

### 5.2. Large-Area Coverage

An important step towards the desired applications was achieved through the discovery that it is possible to coherently modify larger areas by scanning the laser across the surface [76,135,136,137,138,139,140]. In general, there are two scanning methods, as shown in Figure 49: (a) the target is kept in a fixed position and the laser spot is scanned across the sample by means of a pair of orthogonally moving mirrors, controlled by a galvanometric drive (“Galvo scanning head”), whereby the moving beam is focused onto the target by an F-theta (telecentric) lens; (b) the optics are fixed and the target is moved, mounted on a set of precision translation stages.

Both techniques have individual advantages: (a) the procedure allows a very high scanning speed and allows a very compact and rigid setup when using a commercial scanning head; (b) the method allows very large target areas to be covered, depending only on the translation stage size and precision, although it is much slower and generally less compact, meaning it is mostly used for laboratory applications.

Typically, the full surface is covered by the first writing lines along one direction (e.g., “*x*”), followed by adjacent tracks displaced in the other direction (e.g., “*y*”) [139] (Figure 50).

There are two further options, namely writing the lines either along or perpendicular to the laser polarization. Generally, it turns out that scanning normal to the ripples’ direction, i.e., usually along the polarization direction, yields more regular patterns than in the other direction. Another important parameter for the pattern quality is the scanning speed, or more precisely its ratio to the repetition rate, yielding longitudinal pulse overlap and an effective number of pulses acting on one spot or area. A similar role is played by the scanning pitch, i.e., the separation of adjacent tracks (or lateral pulse overlap). The importance of these parameters becomes evident when considering the dose dependence of the generated patterns.

An interesting feature of large-area coverage is presented in Figure 51 [101]. On silicon, the photoluminescence is greatly reduced in the processed area (indicated in Figure 51a) and shown in dark in Figure 51b), reflecting the modulated groove structure. This implies that in this region, the lifetime of photo-induced carriers is strongly reduced by non-radiative recombination. This is a clear sign that the crystalline structure is heavily perturbed by extended defects, most probably dislocations. In contrast, there is not any indication of melting or large-scale amorphization.

However, *well outside* the irradiated area, the photoluminescence is strongly affected (cf. also Figure 14). The surface carrier recombination is, in fact, *coherently continuing* the grove structure scribed in the modified area.

### 5.3. Applications

In the following sections, two typical functionalization approaches are briefly addressed (cf. [141].

#### 5.3.1. Color

The close affinity between LIPSS structures and diffraction by optical gratings has initiated numerous investigations on color modifications using LIPSS. An impressive overview is presented in [132]. Comparable to the angle-dependent multicolor diffraction from a compact disk (CD), a polarization controlled array of differently oriented LIPSS areas, thus, can yield a multicolored picture (Figure 52) [142]. More important than “painting”, however, is the possibility of producing microscopic markings for anti-counterfeit stamping, using only a small area, e.g., hidden in some larger structured field [142,143].

Particular interest has been in the formation of black silicon, notably in order to optimize the spectral absorption for photovoltaic applications. However, this has to be considered with care, since the structural surface modification is associated with the formation of electronic defects, significantly reducing the carrier’s lifetime [101] (cf. Figure 51). Eric Mazur’s group succeeded in overcoming this problem by conducting laser processing under a SF_6_ atmosphere [144] and subsequent annealing at 1200 K to remove hierarchical secondary structures [145].

#### 5.3.2. Wettability

The modification of the surface wettability by LIPSS was first reported in 2006 by the FORTH group on silicon [146] and by Groenendijk and Meijer [147,148], who patterned a stainless steel surface to become super-hydrophobic (cf. Figure 53). Such surfaces could subsequently be used as molds for plastic replicas exhibiting similar hydrophobicity. In fact, is not only possible to make the surface (super-)hydrophobic but hydrophilicity (super-wetting) can also be achieved [132,139], and even surfaces where a water film creeps upwards against gravity can be obtained [149,150].

There have been many approaches since then, showing improvements in hydrophobicity via chemical alkysilane post-treatment [151] and the formation of hierarchical multiscale patterns [152]. Again, it appears that the irradiation dose plays an important role in controlling the functionality [139,146]. There are two aspects that may be considered: (1) the final surface roughness of the LIPSS morphology; (2) chemical changes due to the recovery from instability after processing. Obviously, the roughness depends on the irradiation dose, as demonstrated in Figure 54 [139], where stainless steel was irradiated at a fluence rate of 1.15 J/cm^2^.

Depending on the translation speed, the efficient dose corresponds to *N*_eff_ = 20 pulses/spot (upper row) or *N*_eff_ = 800 pulses/spot (lower row). Whereas the lower dose (upper row) results in regular ripples (Figure 54b) and a relatively smooth surface with roughness *R*_t_ = 0.88 µm (Figure 5c), the higher dose (lower row) shows a hierarchical structure of spikes with ripples on top (Figure 5e) and results in considerably greater roughness of *R*_t_ = 9.54 µm.

At the same time, the chemical composition changed, as indicated in Table 1, which was measured using EDX mapping.

At low roughness (and low oxidation) levels, the stainless steel surface appears to be mostly hydrophobic, while at high roughness (and high oxidation) levels it becomes hydrophilic (35 days after processing). After much longer ripening times (17 months), formerly hydrophilic surfaces become super-hydrophobic, with a contact angle > 145° on a hierarchical surface, with 550 nm ripples on top of 3 µm cones (Figure 54e), as shown in Figure 55 [139].

## Figures and Tables

**Figure 1 nanomaterials-13-00379-f001:**
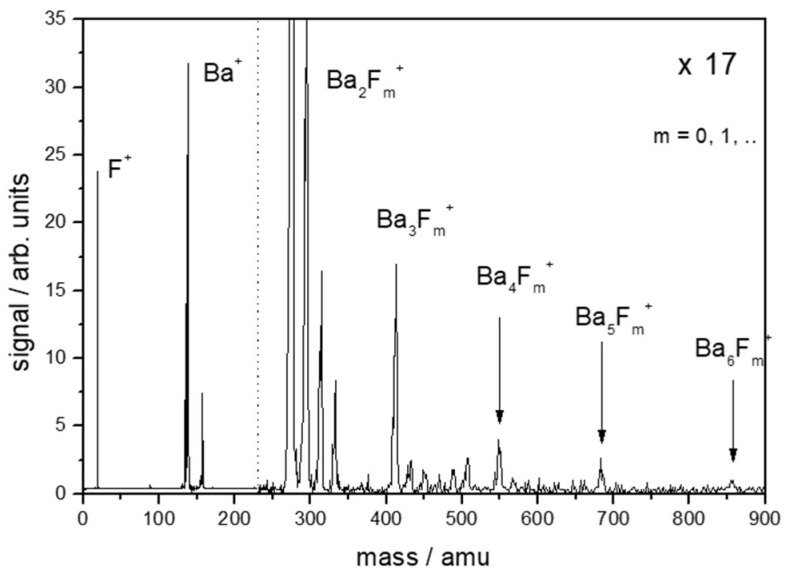
Clustered emission of positive ions (laser desorption) from BaF_2_ after irradiation with ultra-short laser pulses (800 nm, 100 fs) (time-of-flight (T-o-F) mass spectrum; the signal for masses above 250 amu right of the dotted line is magnified by a factor of 17) [31].

**Figure 2 nanomaterials-13-00379-f002:**
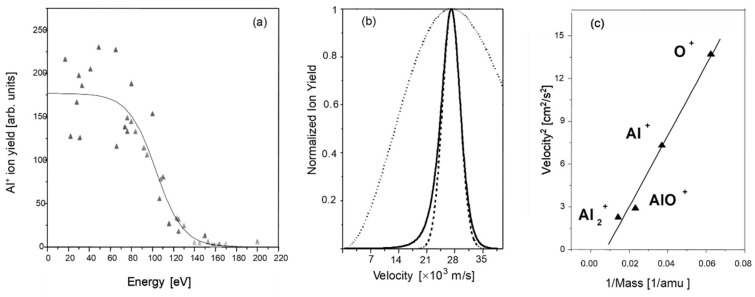
Kinetic energy levels of ions emitted from Al_2_O_3_ [31]: (**a**) retarding potential transmission; (**b**) corresponding velocity distribution (solid line) compared to a Maxwell–Boltzmann distribution (dotted) with the same maximum and a shifted Maxwellian distribution (dashed); (**c**) kinetic energy levels of different desorbed clusters.

**Figure 3 nanomaterials-13-00379-f003:**
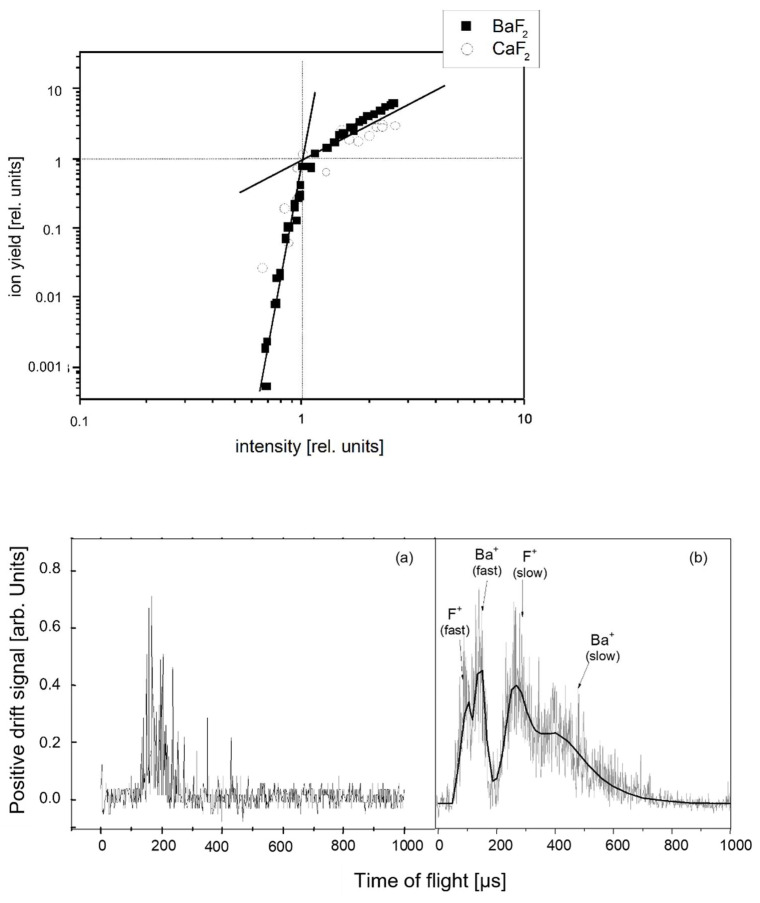
Upper panel: The transition between “desorption (multi-photon)” and “ablation (shielded ion emission)” regimes for two different materials, as indicated. The data are normalized to the transition between pure ion desorption (steep slope of ion yield) and mostly neutral ablation (moderate slope of *ion* yield; neutral ions cannot be seen in the *ion* signal). The transition points are at 0.35 × 10^12^ W/cm^2^ for BaF_2_ and 2 × 10^12^ W/cm^2^ for CaF_2_, respectively. Lower panels: Drift-mode T-o-F spectra from BaF_2_ at intensities (**a**) below and (**b**) above the saturation threshold. The solid line in (**b**) is a fit assuming identical kinetic energies (fast peaks) and temperatures (slow peaks) for both species, Ba^+^ and F^+^. [31].

**Figure 4 nanomaterials-13-00379-f004:**
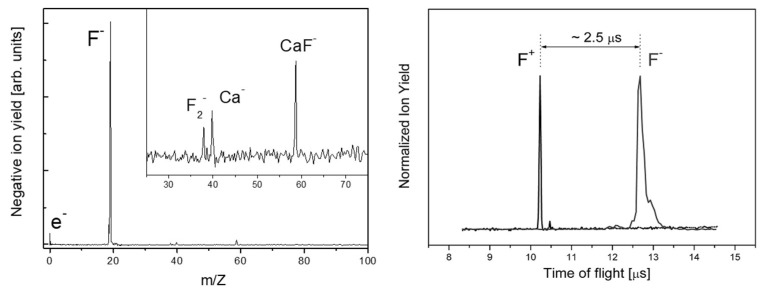
Negative ions observed during ablation from CaF_2_ (**left panel**): T-o-F mass spectrum). As can be seen on the (**right panel**), the negative ions’ distribution is much broader and slower than for the positive ions, indicating a different ablation mechanism. In fact, the negative ions’ arrival time cannot only be explained by the lower drift velocity but also by the later generation time, such as in the ablation plume [31].

**Figure 5 nanomaterials-13-00379-f005:**
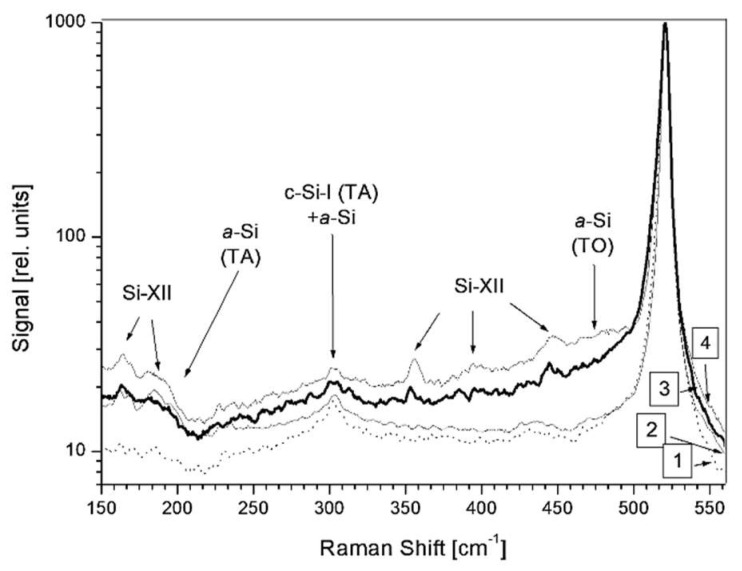
Micro-Raman spectra from the affected area on p-doped Si(100) taken at a depth of several μm [31]. Trace 1 is taken at a virgin area outside the spot, and the others at several positions inside the spot. Due to the penetration depth of the 532 nm laser of ≈1 μm into the bulk, all spectra are dominated by the TO–phonon peak of crystalline silicon at 520.7 cm^−1^. The other peaks correspond to amorphous (a-silicon) or complex tetrahedral (XII) silicon.

**Figure 6 nanomaterials-13-00379-f006:**
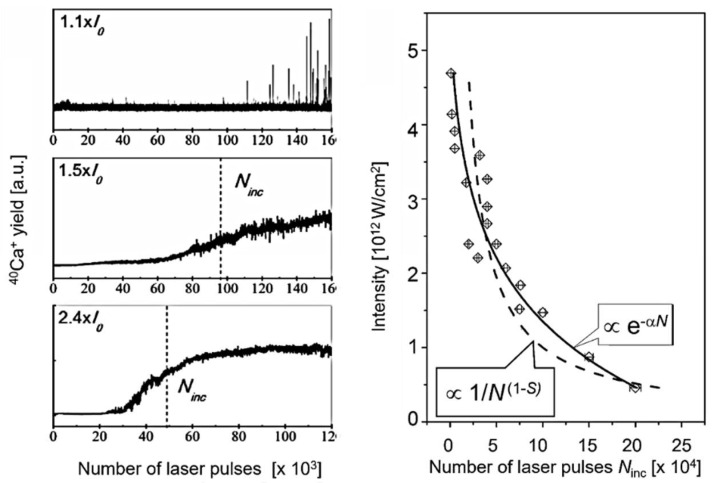
Incubation behavior of Ca^+^ ions from CaF_2_. (**Left**): T-o-F spectra at different laser intensities indicated in the panels (*I_0_* = 1 TW/cm^2^). For each ‘incubation’ plot, the characteristic number of pulses, *N_inc_*, for reaching half of the saturation yield is indicated by a dashed line (this value is defined more clearly than the pulse number for which saturation is reached) [42]. (**Right**): ‘Incubation’ data for Ca^+^ yields, i.e., the dependence of the threshold intensity on the number of ‘incubative’ pulses, *N_inc_*; the solid line exhibits an exponential decay fit to the experimental data, while the dashed line fits the statistical model [43].

**Figure 7 nanomaterials-13-00379-f007:**
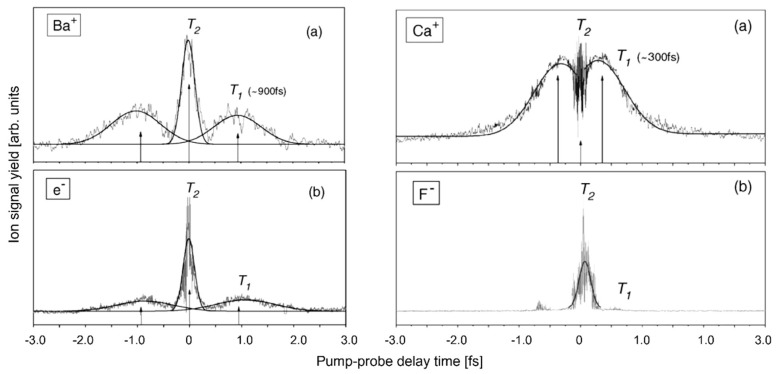
(**Left panels**): Pump probe spectra of (**a**) Ba^+^ and (**b**) electrons from BaF_2_ irradiated by pulse pairs close to the desorption threshold (*I_pump_* ≅ *I_probe_* = 0.5 × 10^12^ W/cm^2^). The solid lines are Gaussian fits to the data, yielding relaxation times of *T2*~250 fs, *T1*~0.9 ps. (**Right panels**): Pump probe spectra of (**a**) Ca^+^ and (**b**) negative F from CaF_2_ irradiated by pulse pairs close to the desorption threshold (*Ipump* ≅ *Iprobe* = 0.9 × 10^12^ W/cm^2^) [56].

**Figure 8 nanomaterials-13-00379-f008:**
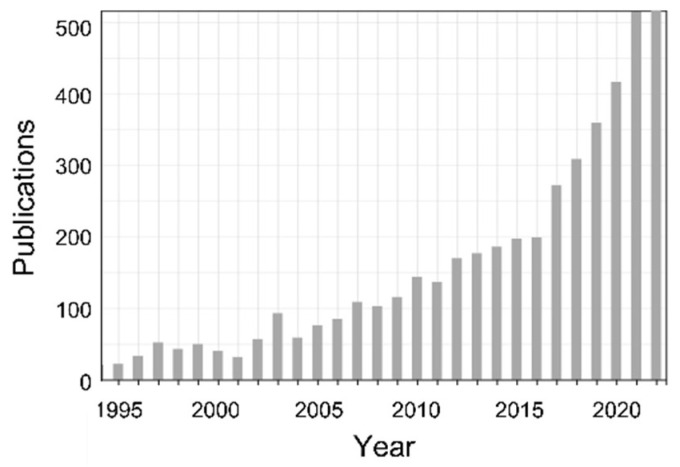
Publications on laser-induced periodic structures (Web of Science, 24 December 2022; thanks to Joern Bonse for eliciting the data). Search topics: “laser-induced periodic surface structures” OR “LIPSS” OR “laser ripples”.

**Figure 9 nanomaterials-13-00379-f009:**
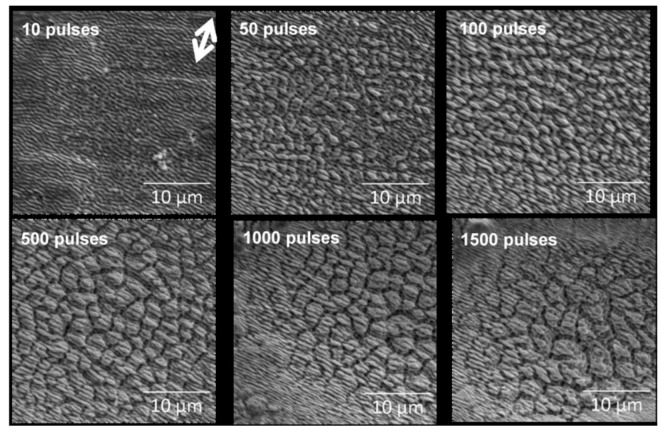
Different feature sizes on laser-irradiated titanium: HSFL, LSFL, grooves, cones [76]. The double arrow on the first panel indicates the laser polarization.

**Figure 10 nanomaterials-13-00379-f010:**
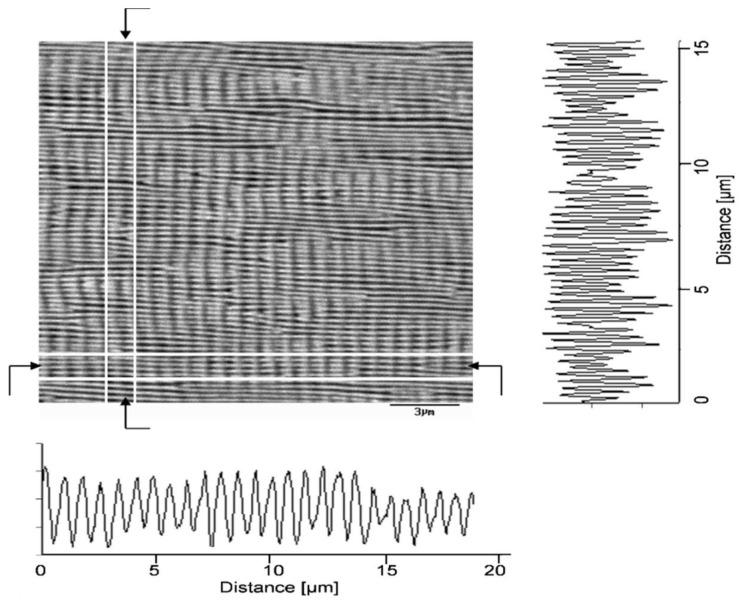
LIPSS formation on BaF_2_ (43,000 laser pulses at 120 fs, 0.9 × 10^13^ W/cm^2^), showing both HSFL (horizontal; cf. profile at the right side) and LSFL (perpendicular; profile shown at the bottom). The laser polarization is parallel to the LSFL, i.e., vertical in the panel [69].

**Figure 11 nanomaterials-13-00379-f011:**
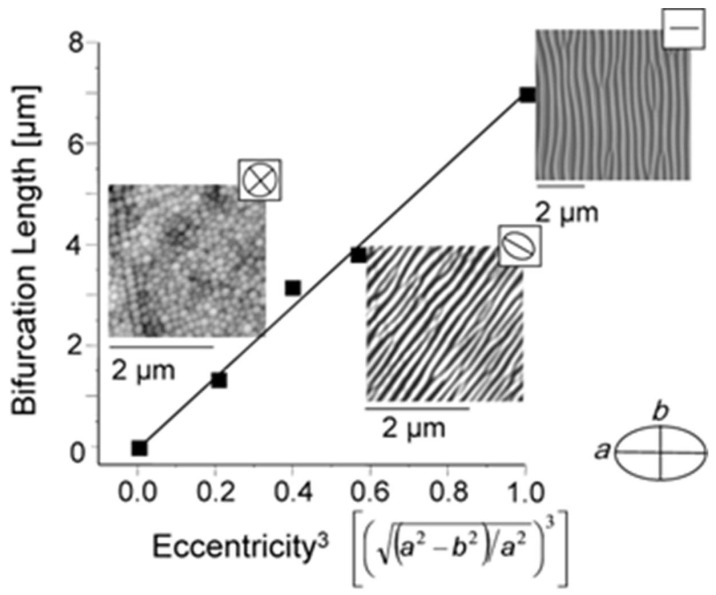
Dependence of LIPSS on the polarization of the incident laser. For linear, elliptical, or circular polarization types, the length of the generated ripplets is proportional to the eccentricity, as given by the semi-axes of the polarization ellipse [62].

**Figure 12 nanomaterials-13-00379-f012:**
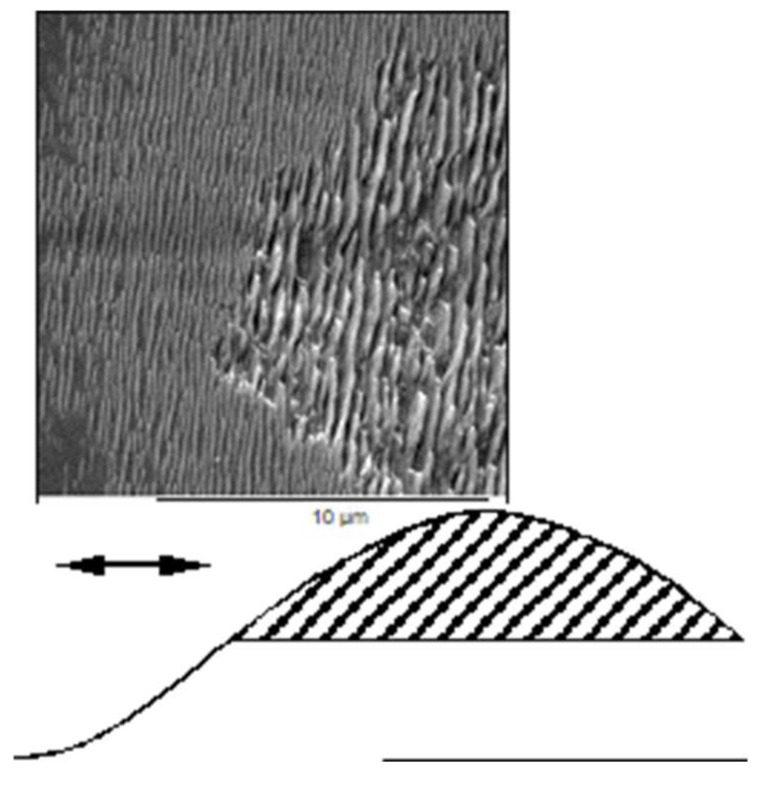
LIPSS formation on one single interaction spot on CaF_2_ (upper), irradiated by 9200 laser pulses (pulse duration = 120 fs, wavelength = 800 nm, total fluence ≈ 1 J/cm^2^). Depending on the local fluence (cf. schematic beam profile below), the spacing, *Λ*, is far below the laser wavelength (*Λ* ≈ 200 nm) at low fluence, and with an abrupt transition it is much larger (*Λ* ≈ 450 nm) in the high fluence center. The double arrow indicates the direction of laser polarization [31,70].

**Figure 13 nanomaterials-13-00379-f013:**
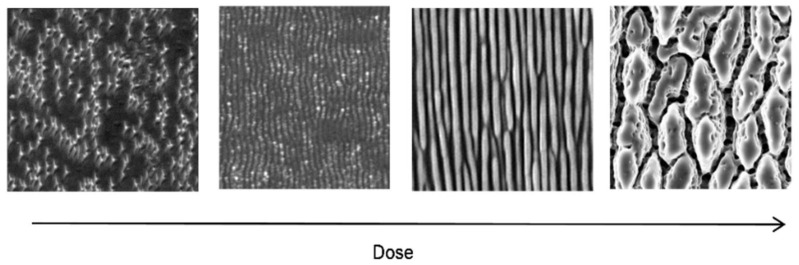
Dose (number of pulses × pulse energy) dependence of LIPSS structures [77].

**Figure 14 nanomaterials-13-00379-f014:**
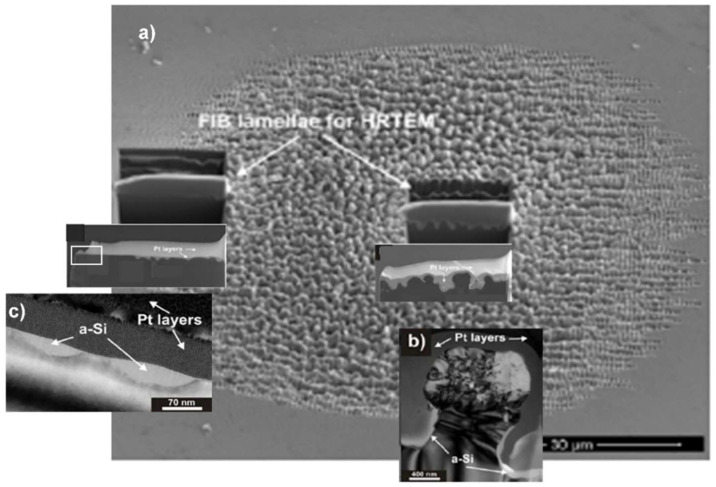
HRTEM investigation of the subsurface structure at the LIPSS spot on Si(100) [80]: (**a**) SEM overview of the irradiated spot, indicating the FIB lamellae; (**b**) cross-section of one ripple at the spot’s center; (**c**) “flat” area outside the modified spot, as indicated by the white frame in the low-resolution inset (**a**).

**Figure 15 nanomaterials-13-00379-f015:**
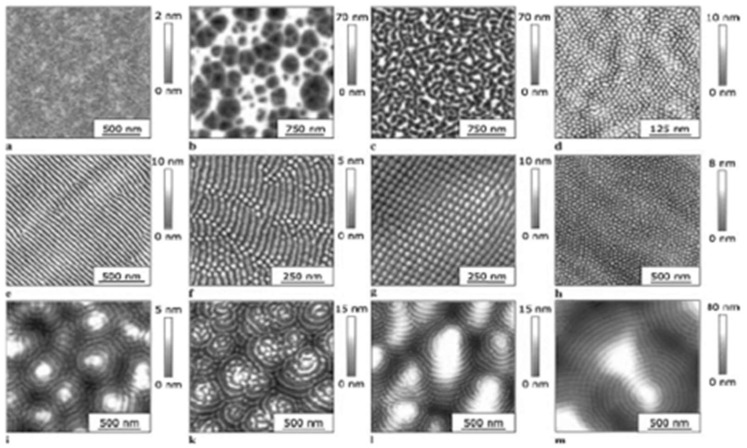
Surface patterns obtained via low-energy ion beam sputtering (from [83]).

**Figure 16 nanomaterials-13-00379-f016:**
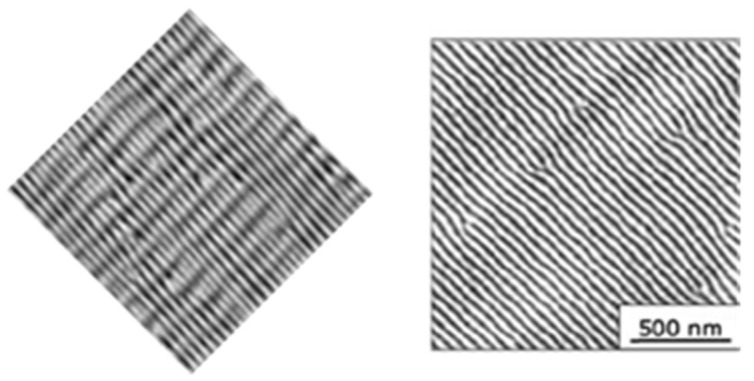
Comparison of LIPSS ((**left**); from [69]) and ion beam surface patterns ((**right**); from [83]); both panels are at the same scale.

**Figure 17 nanomaterials-13-00379-f017:**
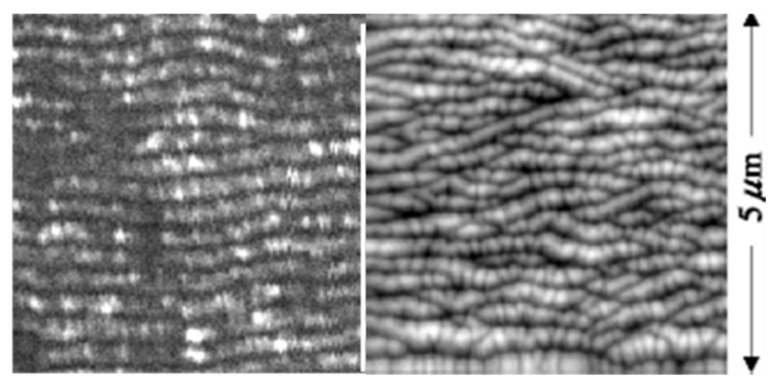
Direct comparison of nanostructures on silicon (at the same scale) from LIPSS ((**left**); from [84]) and ion beam sur face patterns ((**right**); from [85]).

**Figure 18 nanomaterials-13-00379-f018:**
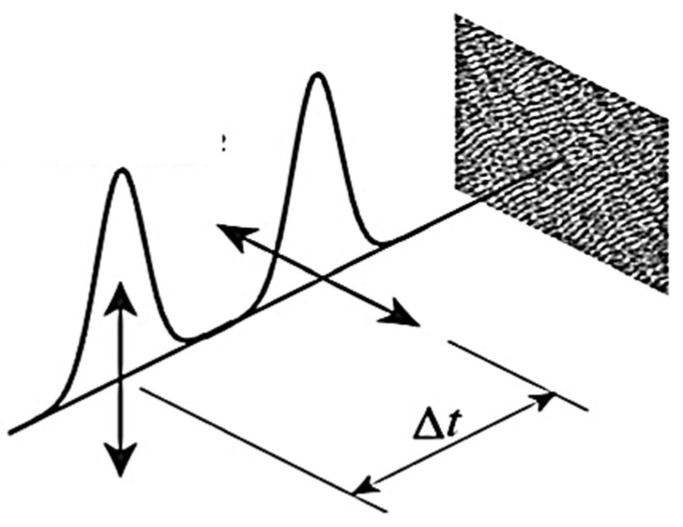
Typical arrangement for double-pulse exposure. Two pulses of identical fluence but orthogonal polarization hit the target with a defined inter-pulse time delay, Δ*t* (from [86]).

**Figure 19 nanomaterials-13-00379-f019:**
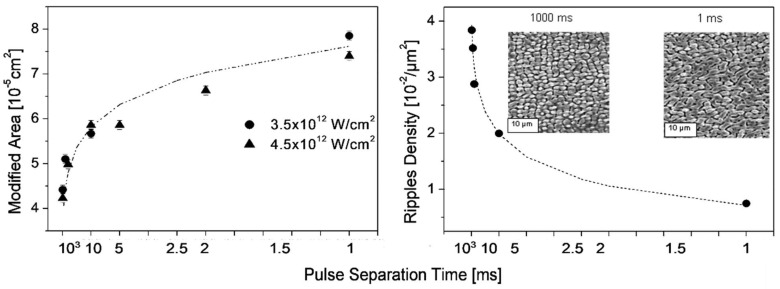
Influence of the pulse-to-pulse separation time on the modified area (**left**) and ripple density (inverse spacing; (**right**)). Silicon irradiation conditions: 1000 pulses with energy of 25 ÷ 30 μJ, (after [78].)

**Figure 20 nanomaterials-13-00379-f020:**
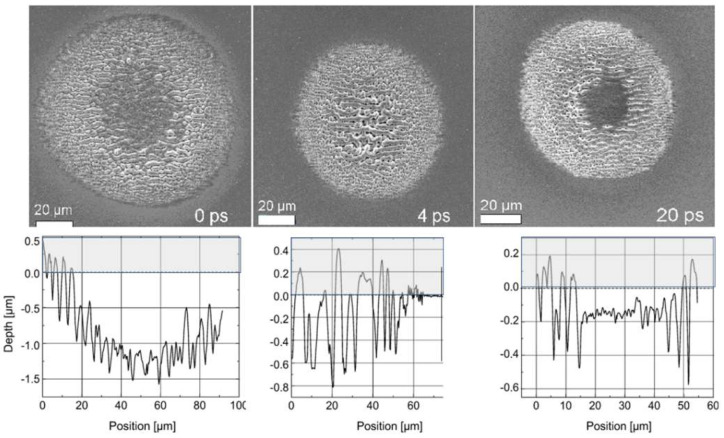
Influence of the double-pulse separation time on the LIPSS structure (after [91]). (**Upper panels**): SEM micrographs (pulse-to-pulse delay indicated). (**Lower panels**): Corresponding AFM traces. The grey shaded area is above the pristine surface (silicon, 50 pulse pairs, Ø 100 μm, 2 × 2.3 TW/cm^2^).

**Figure 21 nanomaterials-13-00379-f021:**
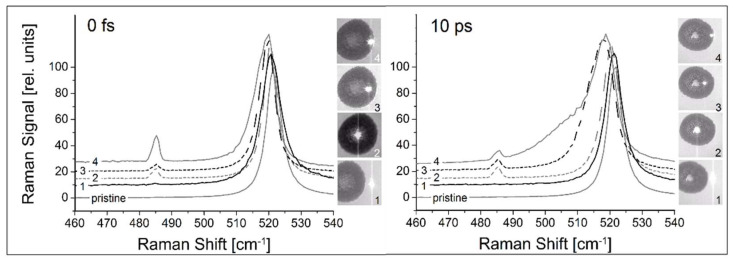
The µ-Raman analysis of two modified areas (0 delay and 10 ps pulse-to-pulse delay) [91]. The different traces in the main panels were taken at the positions indicated by the bright spots in the inserts. For comparison, in each panel the spectrum taken in an untreated (pristine) area of the target is shown.

**Figure 22 nanomaterials-13-00379-f022:**
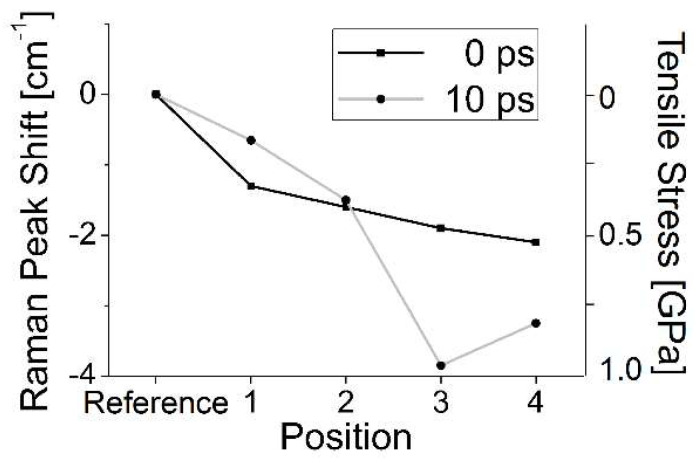
Shift of the TO Raman peak in Figure 21 (the pulse-to-pulse delay times and the position number correspond to Figure 21). Obviously, the induced stress (i.e., lattice deformation) is largest at the crater slope (position 3) for the separated pulses [91].

**Figure 23 nanomaterials-13-00379-f023:**
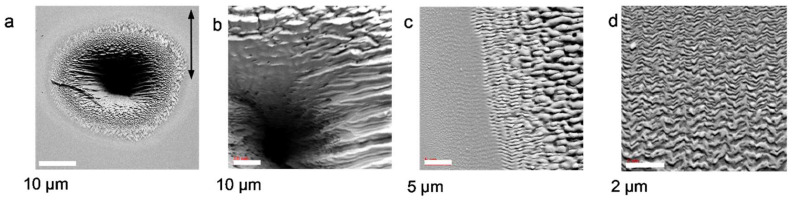
SEM micrographs of an ablated spot on the Si (2.6·× 10^12^ W/cm^2^; 50,000 pulses) [78], where the arrow indicates the direction of laser polarization: (**a**) general overview of the crater; (**b**) details from the center; (**c**) fine ripples at the edge; (**d**) zigzag patterns at the edge or outside.

**Figure 24 nanomaterials-13-00379-f024:**
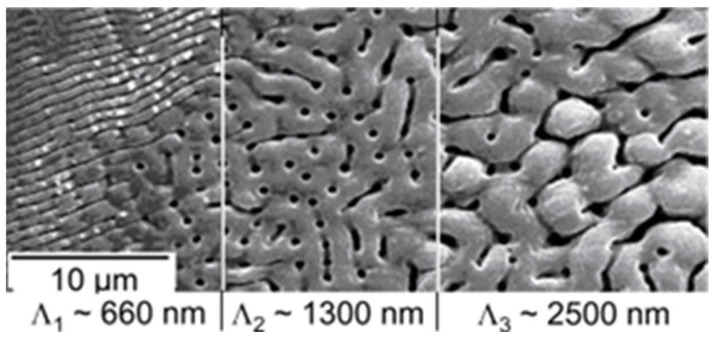
Period doubling of LIPSS structures across one spot on silicon with increasing fluence from left to right (toward the spot’s center). The white lines separating the three areas with different structure widths, *Λ*, are drawn only to guide the eye [99].

**Figure 25 nanomaterials-13-00379-f025:**
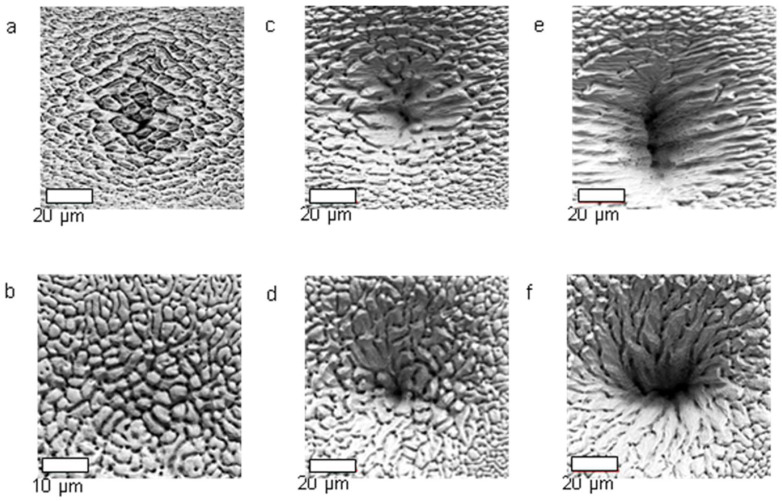
Pattern development at the center of an ablated crater [78] (laser intensity I = 2.6·× 10^12^ W/cm^2^; upper panels: linear polarization; lower panels: circular polarization): (**a**,**b**) 3000 pulses; (**c**,**d**) 5000 pulses; (**e**,**f**) 10,000 pulses.

**Figure 26 nanomaterials-13-00379-f026:**
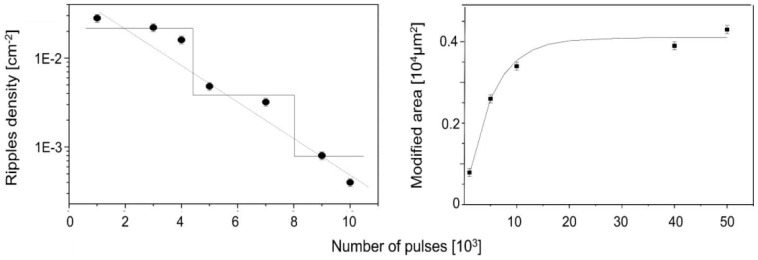
Multi-pulse ablation from silicon @ 2.6 GW/cm^2^; [100] pulse number dependence of the ripple density ((**left**); the “steps” visualize dose-dependent period doubling, cf. Section 4.3) and the modified ablation area ((**right**); the dotted line is a fit to Equation (1)).

**Figure 27 nanomaterials-13-00379-f027:**
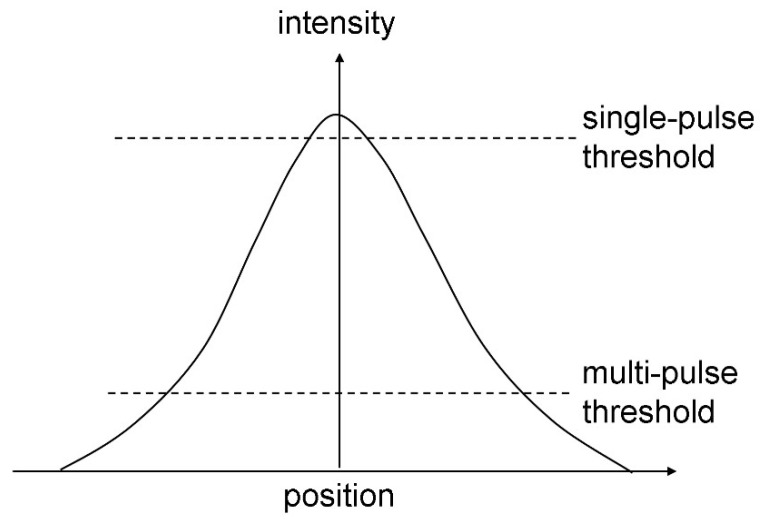
Schematic explanation for the increasing modified area size with increasing doses. The threshold energies for single- and multi-pulse ablation are compared with the spatial pulse profile [101].

**Figure 28 nanomaterials-13-00379-f028:**
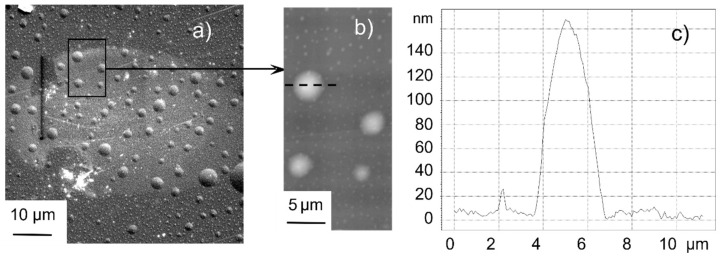
Low dose (100 pulses @ 25% ablation threshold) irradiated spot on CaF_2_ [102]: (**a**) SEM overview of the irradiated spot; (**b**) AFM details of the area indicated in (**a**); (**c**) cross-section along the dashed line in (**b**).

**Figure 29 nanomaterials-13-00379-f029:**
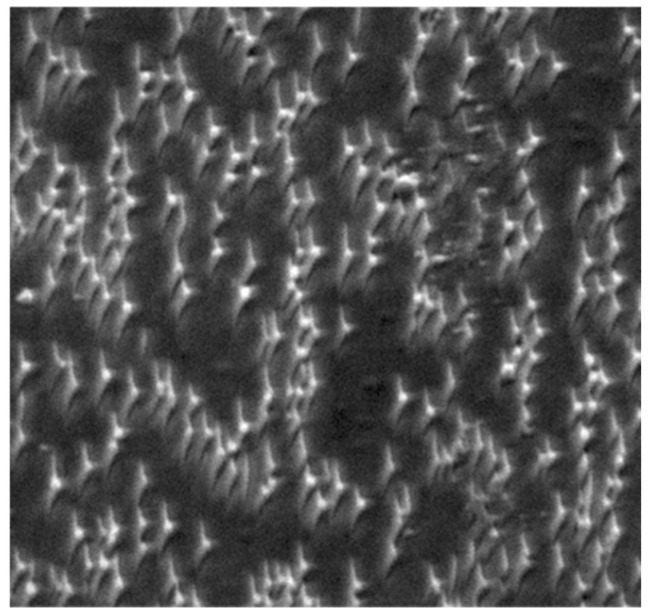
A higher dose (10,000 pulses @ 25% ablation threshold) irradiated spot on CaF_2_ [102].

**Figure 30 nanomaterials-13-00379-f030:**
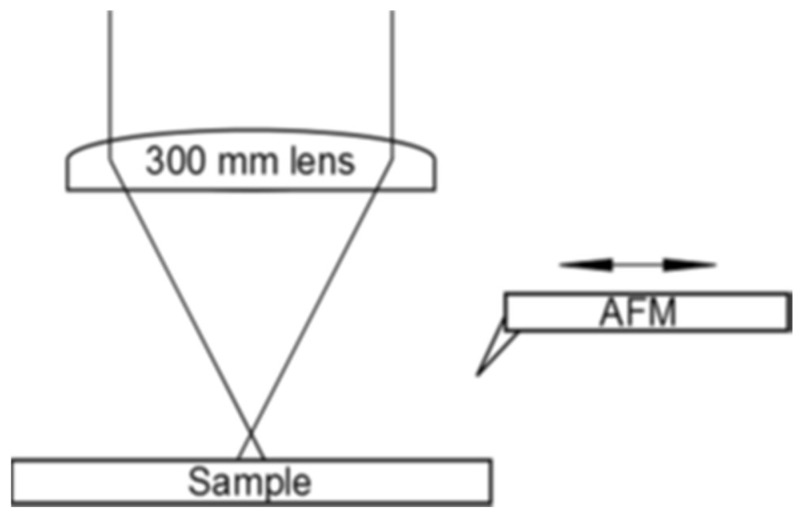
Schematic arrangement for *single-spot evolution* experiments [103].

**Figure 31 nanomaterials-13-00379-f031:**
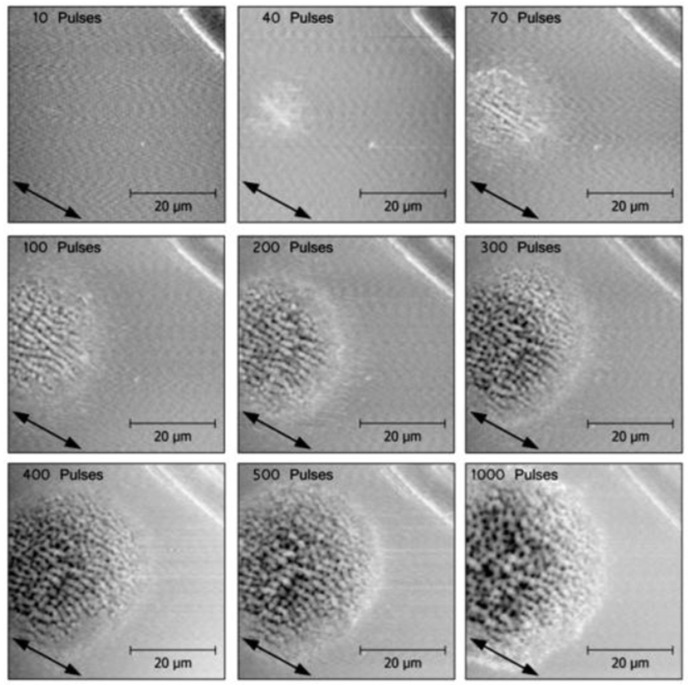
*Single* spot on Si, irradiated with increasing numbers of pulses (indicated on the panel tops) at ≈50% of the ablation threshold fluence. The double arrow indicates the laser polarization; in the upper right corner of each panel, the stitching mark is shown.

**Figure 32 nanomaterials-13-00379-f032:**
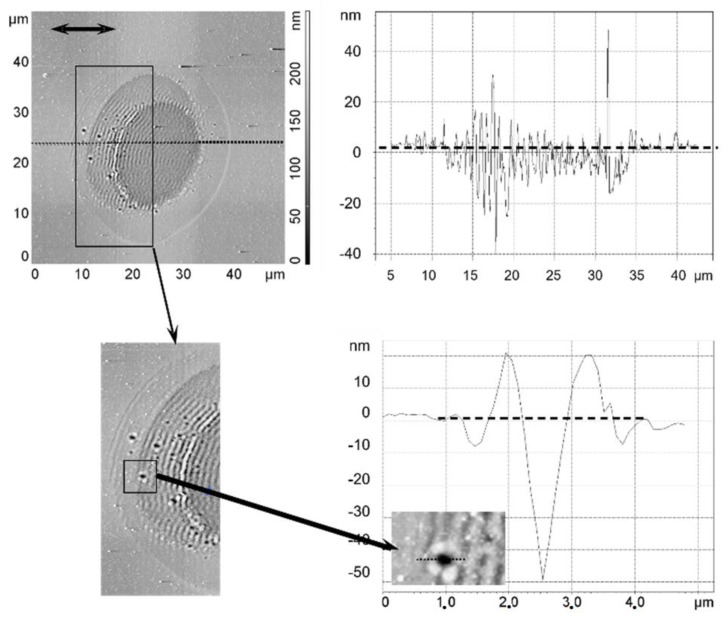
Single-pulse irradiation of silicon at the approximate ablation threshold [103]. (**Upper panels**): AFM overview (**left**) and cross-section (**right**). (**Lower panels**): Details (cf. marked areas) and cross-section around a “hole” at the edge. The dashed lines in the cross-sections denote the pristine surface level.

**Figure 33 nanomaterials-13-00379-f033:**
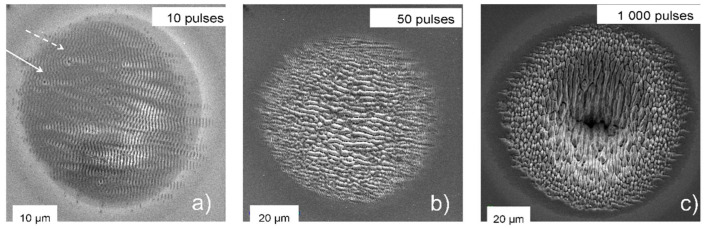
Evolution of irradiated spot morphology on silicon with increasing doses (number of pulses at 2.15 GW/cm^2^ at the spot’s center, close to the ablation threshold). Panels (**a**–**c**) show spots with different irradiation (number of pulses indicated in the upper right corner of the panels) The white arrows in panel (**a**) point to two “hole” defects, as shown in detail in Figure 34 [77,103].

**Figure 34 nanomaterials-13-00379-f034:**
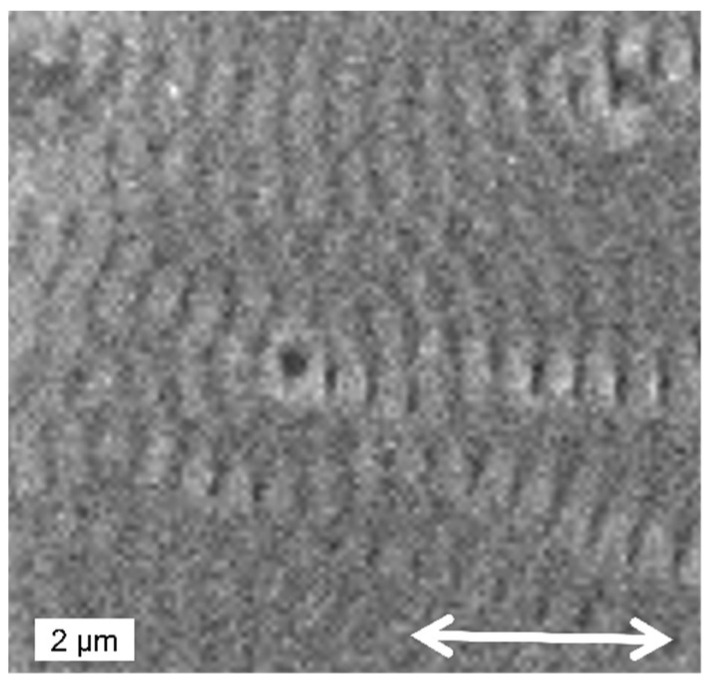
Detail of the “hole” (white arrow in Figure 33a) in the outer region of the low-dose irradiated spot, surrounded by a concentric ripple pattern. [77]. (The white double arrow indicates the laser polarization).

**Figure 35 nanomaterials-13-00379-f035:**
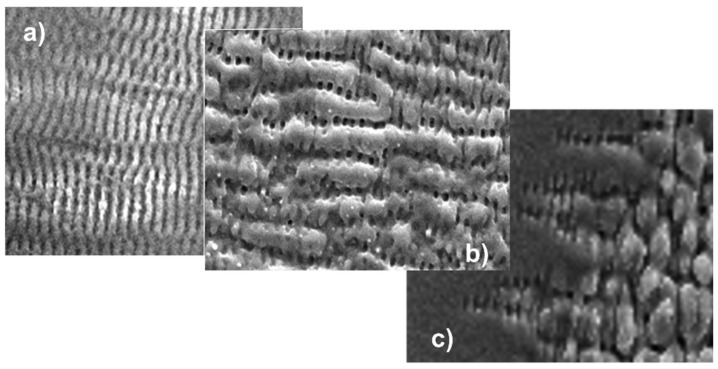
Magnified details (same scale) of the irradiated spots in Figure 33 [77]: (**a**–**c**) images corresponding to the respective spots in Figure 33.

**Figure 36 nanomaterials-13-00379-f036:**
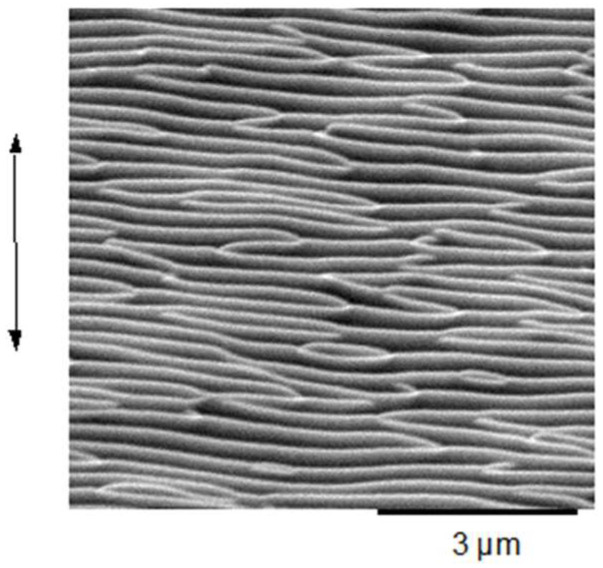
Bifurcations on CaF_2_ (5 × 10^3^ pulses at 5 GW/cm^2^) [107].

**Figure 37 nanomaterials-13-00379-f037:**
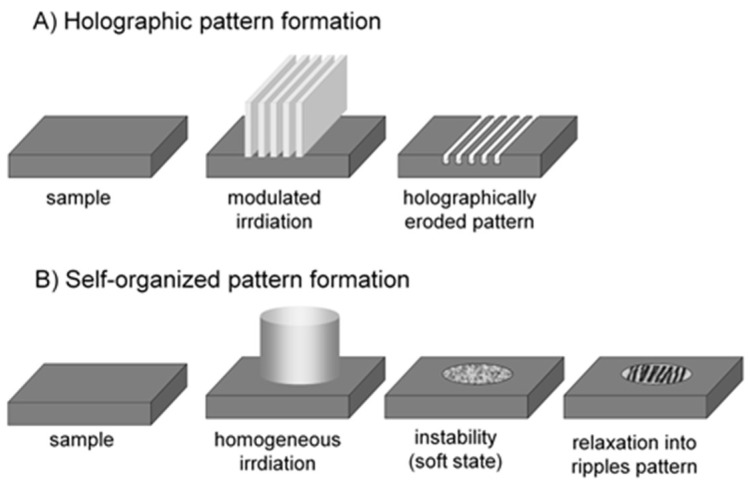
Two principal pathways to LIPSS formation [77]: (**A**) holographic or lithographic ablation reproducing the electromagnetic field distribution; (**B**) self-organized pattern formation, based on the laser-induced creation of surface instability.

**Figure 38 nanomaterials-13-00379-f038:**
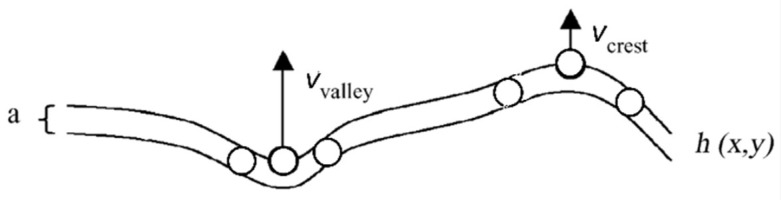
Surface model of a thin, corrugated, liquid-like film [113]. Due to the higher next-neighbor density in the valley, the escape velocity, v, is higher than at the crest. On the other hand, the surface tension strives to minimize the surface and reduce the corrugation via atomic diffusion.

**Figure 39 nanomaterials-13-00379-f039:**
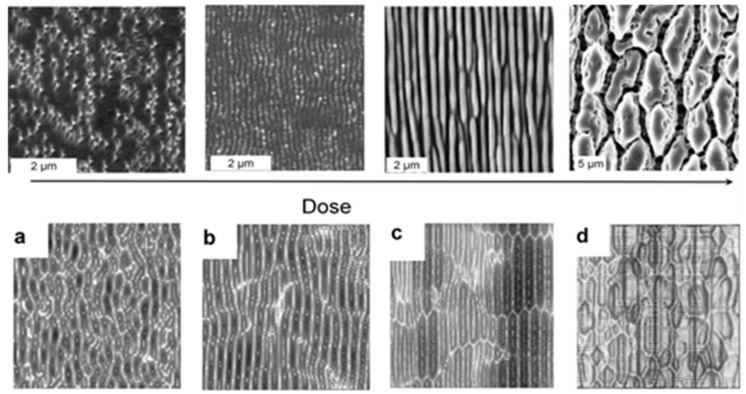
Changes of laser-induced surface patterns with increasing irradiation doses [77]. Upper row: Experiments (SEM micrographs) with increasing numbers of pulses and fluence levels (cf. Figure 13). Lower row: Numerical simulations with increasing durations of instability (time evolution): (**a**) *t* ≪ *t*_c_; (**b**) *t* = *t*_c_/2; (**c**) *t* = *t*_c_; (**d**) *t* = 2*t*_c_. Note that for the simulation, NO REALISTIC scale of *t* can be given because of the lack of detailed material parameters.

**Figure 40 nanomaterials-13-00379-f040:**
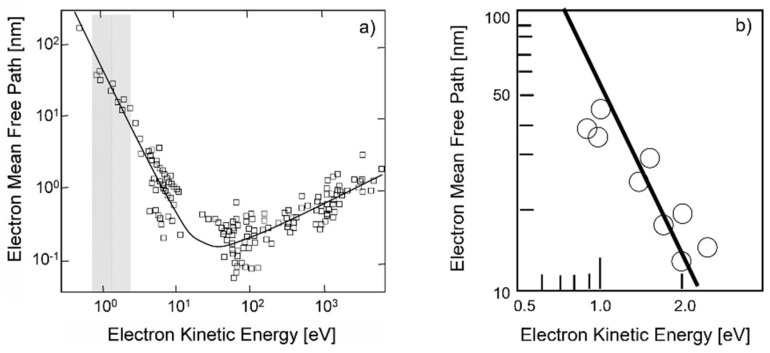
“Universal curve” of the electron mean free path vs. electron kinetic energy [115]. The shaded part (**a**) indicates the region of relevant kinetic energy of conduction band electrons (before escaping the target surface, cf. Section 3.2.1). (**b**) Details of the shaded area in (**a**) are also shown.

**Figure 41 nanomaterials-13-00379-f041:**
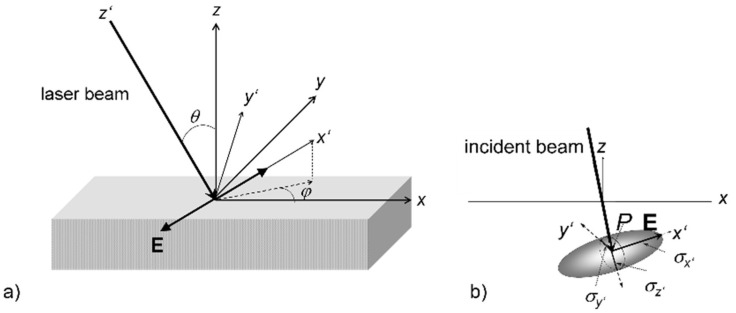
Geometrical situation at the target [117]. (**a**) General overview of laboratory (*x*,*y*,*z*) and laser beam (*x*′,*y*′,*z*′) coordinate systems. The primed laser system is defined by the laser propagation (−*z*′) and polarization (*x*′ parallel **E**). Angles *θ* (incidence) and *φ* denote the azimuthal and polar rotation of the primed laser coordinate system, respectively. (**b**) Detailed situation at the absorbing point *P*. The ellipsoid with axes *σ_x_*, σ*_y_*, σ*_z_* denotes the relative collisional energy transfer cross-sections after laser excitation.

**Figure 42 nanomaterials-13-00379-f042:**
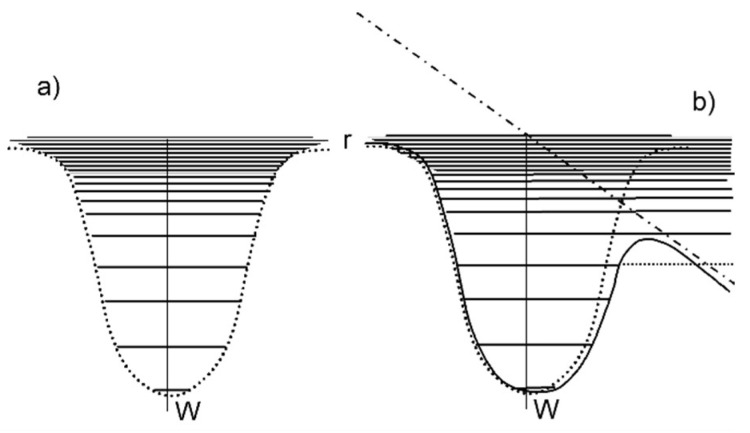
Influence of the laser’s electric field on the atomic Coulomb potential [31]: (**a**) symmetric potential without the laser field; (**b**) deformed potential due to the direction of the laser field (dotted straight line), resulting in an increase in the escaping electrons’ kinetic energy. In a similar way, the field accelerates the electrons in the conduction band.

**Figure 43 nanomaterials-13-00379-f043:**
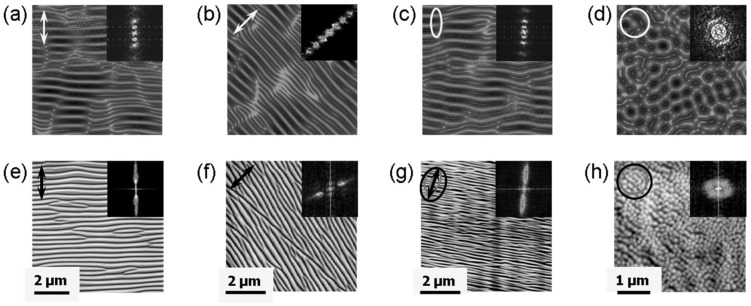
Comparison of polarization-dependent LIPSS between numerically calculated structures ((**upper row**), panels (**a**–**d**)) and experimental SEM micrographs of a CaF_2_ surface irradiated by 5000 pulses at 8 GW/cm^2^ ((**lower row**), panels (**e**–**h**)). The polarization state is indicated by the white and black arrows. The inserts present corresponding 2D-FFT images of the structures [116].

**Figure 44 nanomaterials-13-00379-f044:**
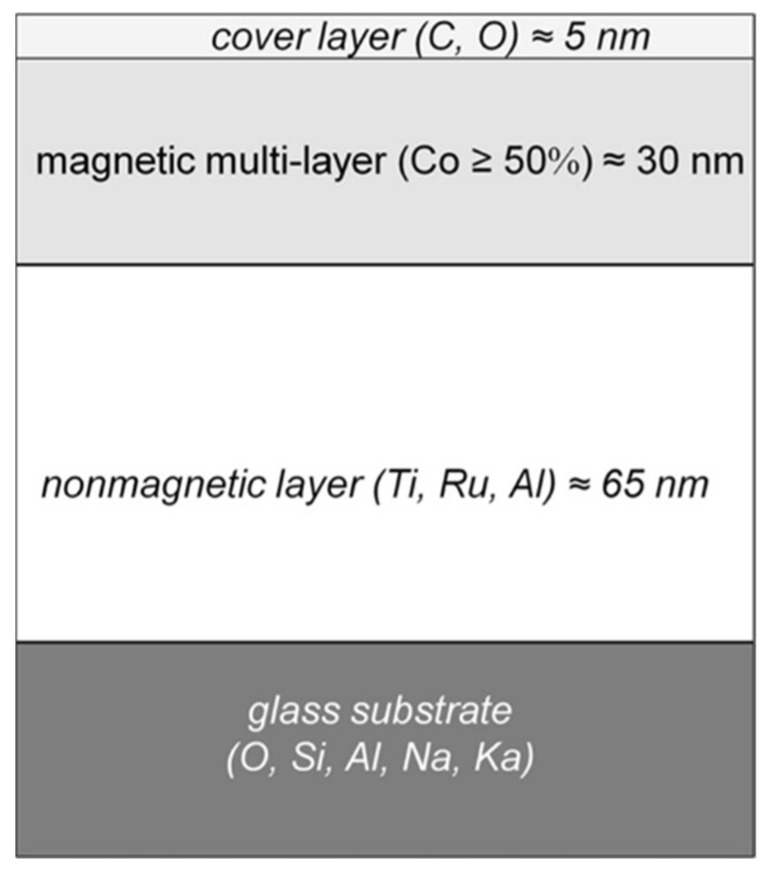
Structure of the magnetic multi-layer target (from depth-resolved EDX spectra) [123] in a computer hard disk, consisting of a magnetic multilayer stack (Co-Pt-Co) of about 30 nm in thickness on a glass substrate with an intermediate (non-magnetic) metallic buffer and covered with a thin polymer film.

**Figure 45 nanomaterials-13-00379-f045:**
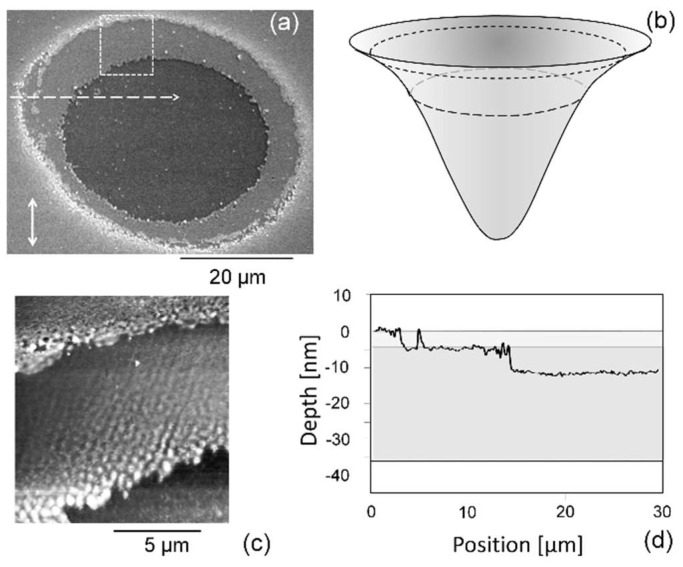
A spot on a complex multi-layer target (cf. Figure 44) [124], irradiated by a single pulse at 0.26 J/cm^2^. (**a**) SEM graph of the irradiated area. The double arrow in the lower left corner indicates the laser polarization. (**b**) Gaussian beam profile, where the dashed lines indicate the borders of the ring structure in (**a**). (**c**) AFM details of the area indicated in (**a**) by a white dotted box. (**d**) AFM profile along the white dashed line in (**a**).

**Figure 46 nanomaterials-13-00379-f046:**
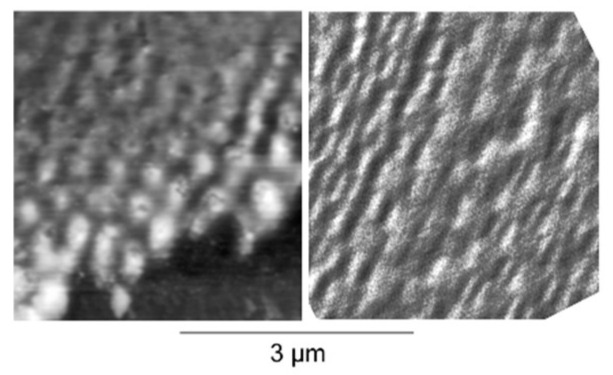
Comparison between the magnetic layer surface morphology (Figure 45c) after a single pulse (AFM, (**left**)) and a magnetic force micrograph (MFM, (**right**)) from an untreated area of the same hard disk [124].

**Figure 47 nanomaterials-13-00379-f047:**
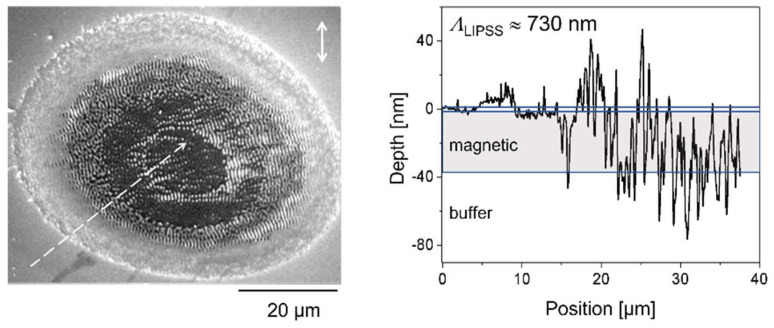
Modified area (cf. Figure 45) after ten pulses at 0.26 J/cm^2^ (after [124]). (**Left**): SEM micrograph, where the white double arrow indicates the polarization. (**Right**): AFM profile along the dashed line in the left panel. Note the swelling [104] above the pristine surface (level “0”).

**Figure 48 nanomaterials-13-00379-f048:**
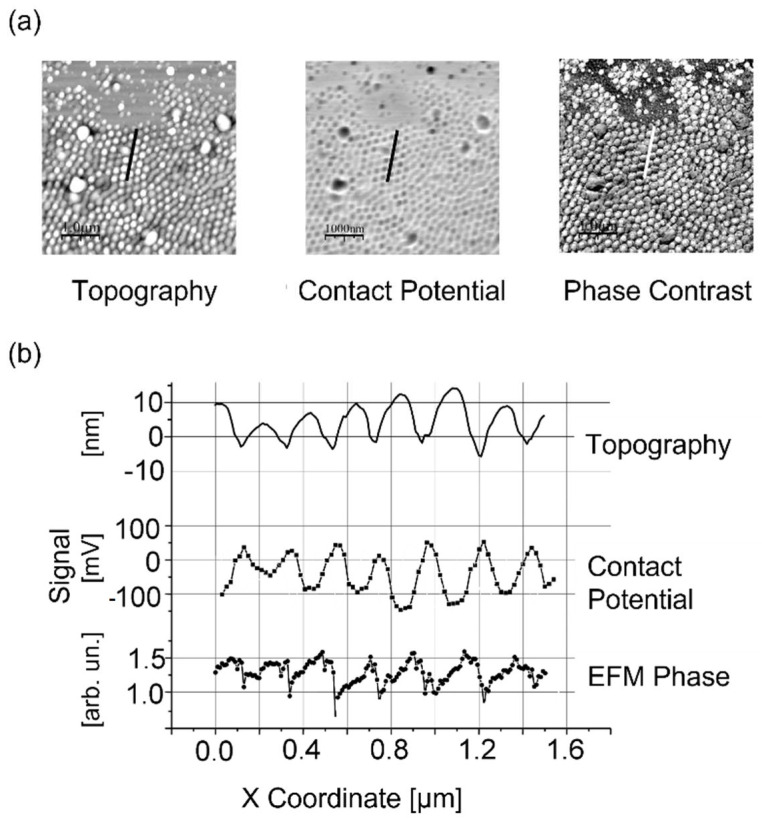
Electrical properties of self-organized nanostructures on p-Si, irradiated by circular polarized light [135]. (**a**) Scanning probe micrographs: AFM (topography), SKM (contact potential), and EFM (phase contrast). (**b**) Corresponding sectional traces along the lines indicated in (**a**). The contact potential is in anti-phase to the topography; the phase contrast corresponds to the derivative of the topography trace.

**Figure 49 nanomaterials-13-00379-f049:**
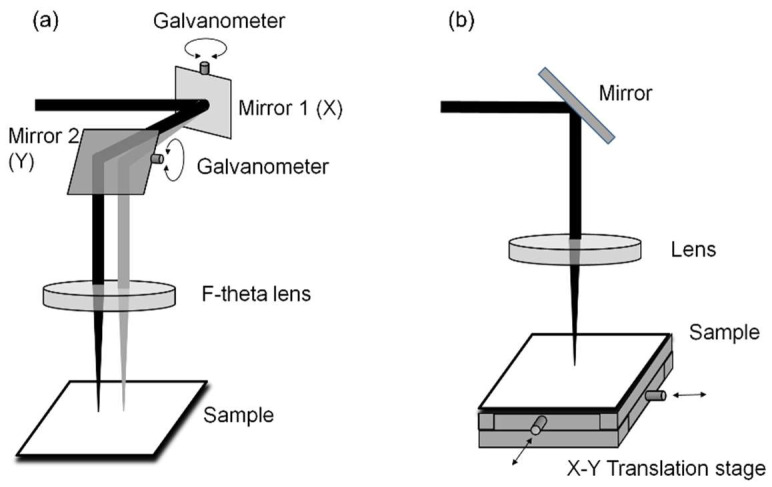
Schematic of two methods used for large-area scanning: (**a**) galvo scanning head and fixed sample; (**b**) fixed laser spot and mechanically scanned sample [130].

**Figure 50 nanomaterials-13-00379-f050:**
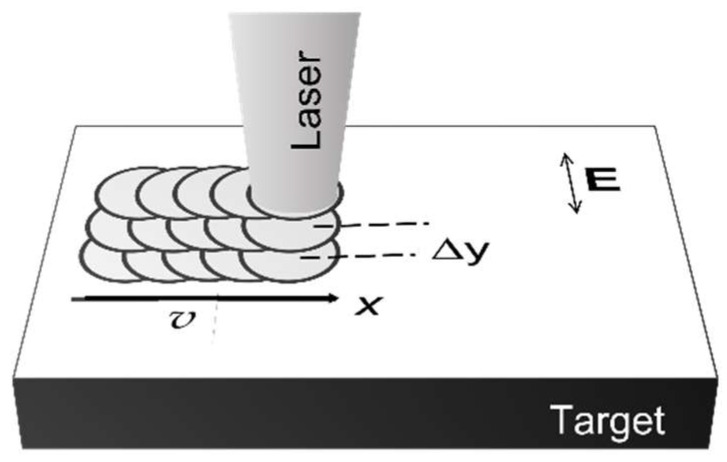
Schematic of the surface coverage (after [139]). First, lines are written in the *x*-direction with speed *v*, then the target or spot is displaced by Δy in the *y*-direction and a new line is written in the *x*-direction, and so on.

**Figure 51 nanomaterials-13-00379-f051:**
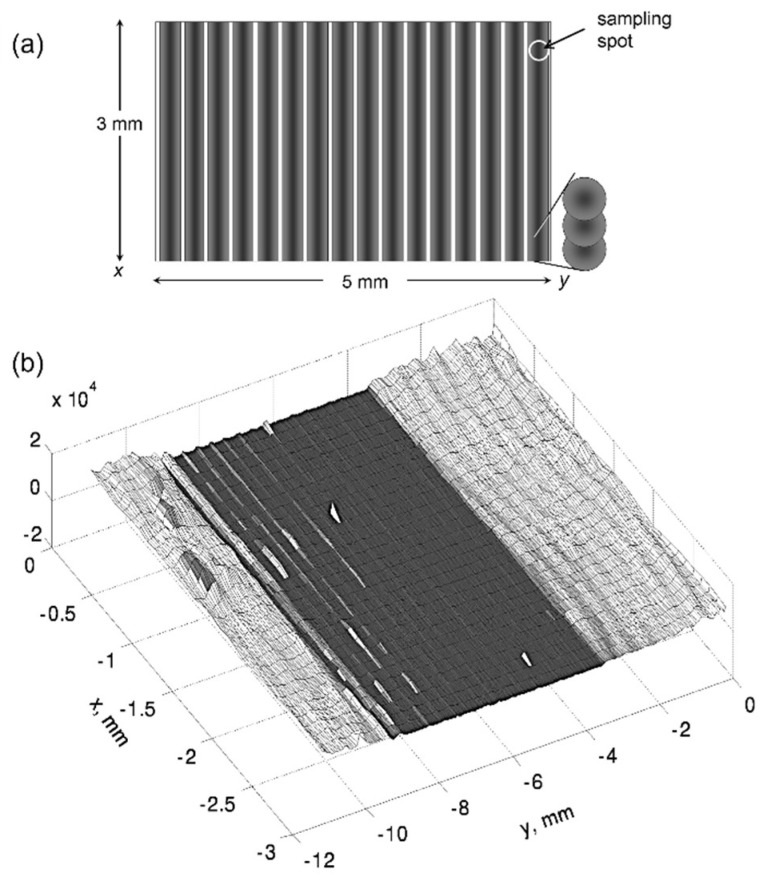
Photoluminescence mapping of a large LIPSS processed area (3 mm × 5 mm) on silicon [101]: (**a**) schematic of the irradiated area; (**b**) photoluminescence map obtained by scanning the PL sampling spot, indicated (**a**) over an area 3 mm × 12 mm.

**Figure 52 nanomaterials-13-00379-f052:**
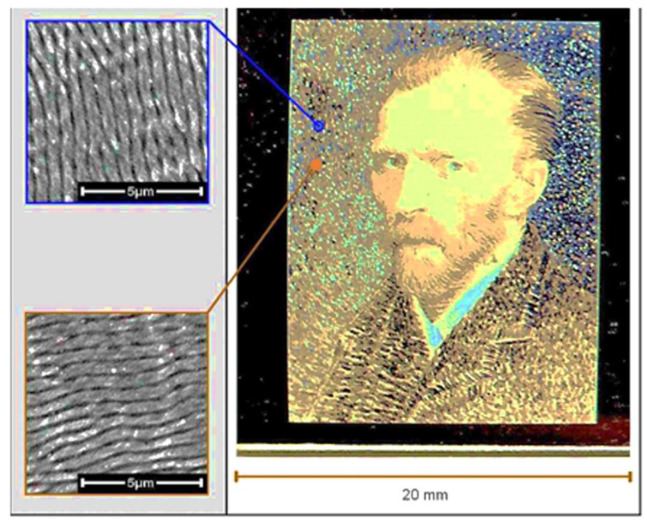
Multi-color diffraction from a laser-structured stainless steel surface with areas of different LIPSS orientation (from [142]). The panels on the left present the respective LIPSS patterns at the positions indicated by the arrows., representing “blue” (**upper**) and “orange” (**lower**).

**Figure 53 nanomaterials-13-00379-f053:**
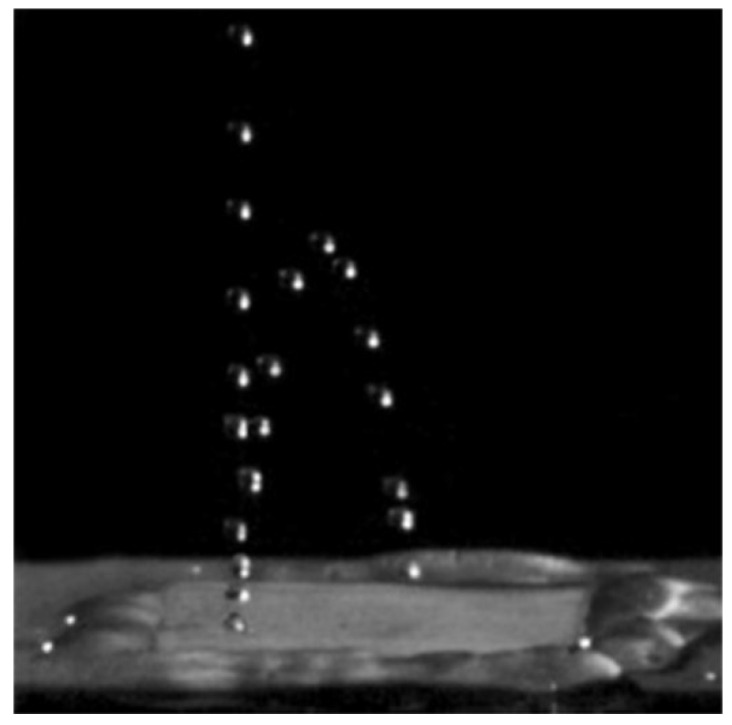
Super-hydrophobicity: falling droplets bounce back from an LIPSS’ surface (from [148]).

**Figure 54 nanomaterials-13-00379-f054:**
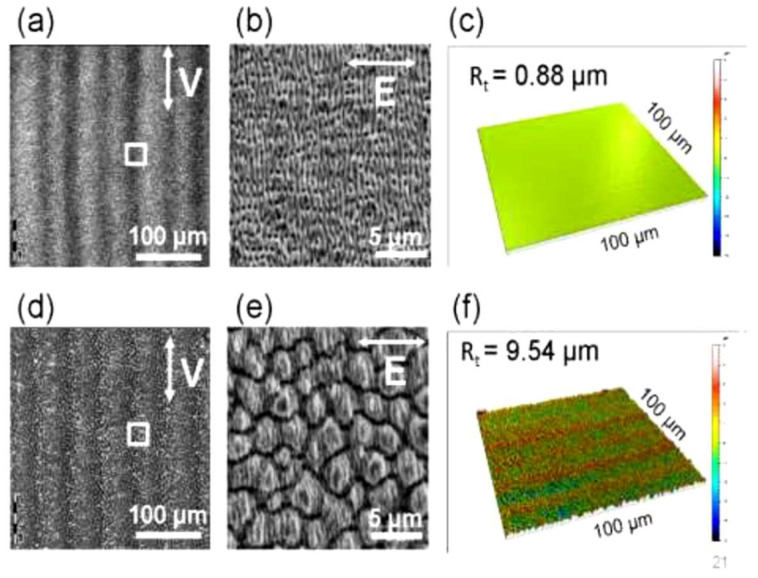
Dependence of LIPSS patterns and surface roughness on the irradiation dose (stainless steel): (**a**–**c**) Neff = 20 pulses/spot; (**d**–**f**) Neff = 800 pulses/spot. Vertical rows: overview (**left**), detailed LIPSS pattern (**middle**), roughness (**right**) [139].

**Figure 55 nanomaterials-13-00379-f055:**
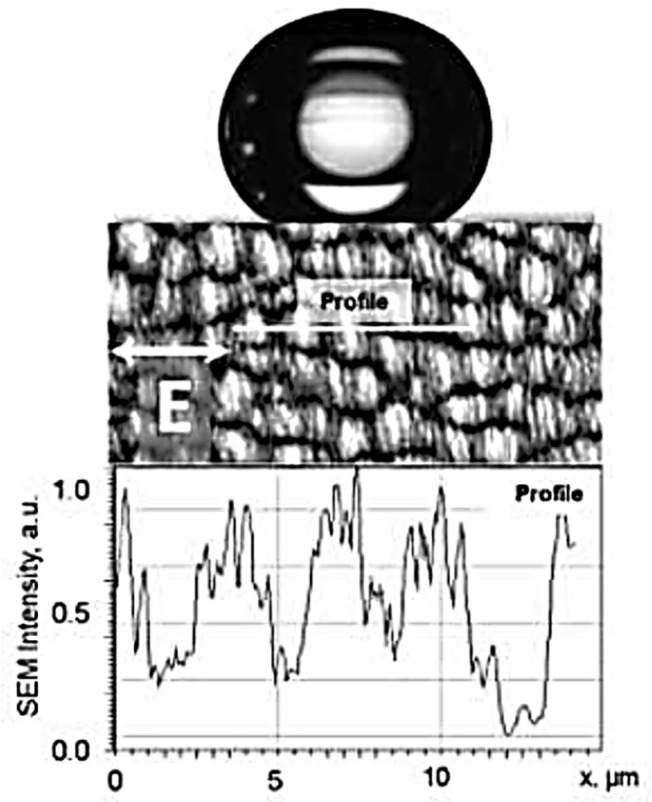
Water droplet on a hierarchically structured 17-month-old stainless steel surface (the middle panel shows an SEM micrograph, while the lower panel shows a cross-section of the hierarchical surface) [139].

**Table 1 nanomaterials-13-00379-t001:** Elemental composition of the stainless steel surface (in %) dependent on the effective irradiation dose [139].

Chemical Elements (%)	O	C	Fe	Cr	Ni
Irradiation Dose					
Non irradiated	0.19	1.79	71.34	17.10	8.12
20 pulses/spot	1.79	3.25	69.30	16.46	7.60
200 pulses/spot	4.92	4.41	65.90	15.66	7.48

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
