# Peer review of "Dynamics and Processes on Laser-Irradiated Surfaces"

_nanomaterials, 2023, doi:10.3390/nano13030379_

Round 1

Reviewer 1 Report

The manuscript mainly reviews the solid surface modification by intense laser pulses and the dynamics of the related process. The paper can be considered to be published after the following concerns are well addressed.

1. The references are not quite updated, especially in the last decade. Please check the recent publications which are related to this topic. For example, some groups use UV laser to excite semiconductors to modify the surface structure with metal decorating, including, but not limited to, ACS Nano 2021, 15, 2, 2947–2961; ACS Nano 202115 (9) , 15328-15341.

2. For the Figures in the manuscript, the image quality of some figures needs improvement. Multiple Figures in one Figure need to be labeled as a, b or c... The labels should be consistent with those in other Figures. For example, it is "(a)" in Figure 2 but changed to "a" in Figure 23, and also "a)" in Figure 28, and "A)" in Figure 37. The unit of the scale should be labeled on the top of the bar. 

3. The format of the references needs significant improvement. Please notice that PHYSICAL REVIEW A is abbreviated as Phys. Rev. A, instead of Re5.

Author Response

A first, general remark to all reviewers:

I well agree with the remarks of all Reviewers that the manuscript is  not reflecting the up-to-date situation in terms of the very vast works in the field, having appeared during the last years (since about 2018), nor is it exhaustive in works before that time. In particular, there are only very few applications of LIPSS, compared to the very large range of publications.

However, the intention for writing the present review should be considered:

On the occasion of the Nanomaterials’ Special issue “Dynamics and Processes at Laser-Irradiated Surfaces—A Themed Issue in Honor of the 70th Birthday of Professor Jürgen Reif“, I was asked to contribute an extended review – which, I understood, should be mostly a compilation of OUR contributions to the field, backed by references which I consider as relevant for our work. The second limitation was set by my retirement in 2018, after which I was somehow restricted in following the evolution of the field in full extension. Also, in terms of applications, I limited the work also to fields where we provided actual contributions.

Detailed response to Reviewer 1:

I wonder about the request "Extensive editing of English ..." since the Reviewer, further down, concedes "correct and readable use of English" (4/5) and all other Reviewers confirm that "English language and style are fine". Therefore, I stay with my manuscript.

#1 References: see general remark, above

#2 Figures: see below "list of changes". For several figures, higher qualtity images are provided. Since all figures are taken from previous manuscripts (no raw data available), changing of labeling was not always possible

#3 Particular bug (e.g., "Re5"). This bug occured during submission and processing: in the References (as well as in the text) the letter "v" was wrongly "translated" into the number "5" (cf. Latin number "v"=5). The bug was corrected throughout the manuscript.

List of changes:

Figs. 1-4,, 7: replaced by figs. with higher resolution

Figs. 5, 11, 22: shifted in position

Fig. 8: improved legend

Fig.10: corrected deformation of axes (X,Y same scale)

Fig 21: the figure had been lost from the manuscript during submission/processing; figure newly introduced into the manuscript

Several incomplete, misleading or bad formulations corrected

Typographical errors corrected

Ref.[98]: Title of publication added

Reviewer 2 Report

The manuscript presents a well-written review of works on laser surface treatment of a number of crystals and metals, with the main emphasis on the results on Laser Induced Periodic Surface Structures (LIPSS), which, together with the list of publications, make up ¾ of the entire review. The review is written in sufficient detail, competently and accessible to a wide range of readers. It is pleasant that, unlike some other similar reviews, the main emphasis in the Juergen Reif’s review is on the description of the results obtained, and not on the analysis of bibliographic data statistics. The review organically combines theoretical concepts and experimental data. It is impossible to cover all the topics of laser surface treatment of materials in one review, but I would like to express my opinion on the revision of the text of the review. Unfortunately, many modern works are missing from the bibliography. Only six publications from the list of references refer to the period after 2018. Perhaps that is why the text of the review does not include important theoretical works by H. Zhang and A. Rudenko. The complete list of materials for which experimental data are presented in the manuscript includes BaF2, CaF2, silicon, titanium, stainless steel and magnetic multilayer layer (Co-Pt-Co). In my opinion, works on laser surface treatment of diamond and a wide class of carbon films, in which laser radiation induces a diamond-graphite phase transition, deserve at least a brief mention in the text of the review. In recent years, work on laser surface treatment has been very actively carried out by Chinese researchers, but this was not adequately reflected in the text of the review. The areas of application of LIPSS in the review are rather limited. For example, the use of LIPSS for surface enhanced Raman scattering (SERS) techniques is not mentioned. It would not be superfluous in the list of references to refer to other reviews on related topics, for example, to B. Reillard, F. Mucklich. Ablation effects of femtosecond laser functionalization on surfaces. Chapter 24 (and other Chapters) in Laser Surface Engineering, Processes and Applications. Edited by: J. Lawrence and D.G. Waugh. Woodhead Publishing Series in Electronic and Optical Materials (2015) and on L. Cerami, et al, Femtosecond Laser Micromachining, Chapter 12 in R. Thomson et al. (eds.), Ultrafast Nonlinear Optics, Scottish Graduate Series, Springer (2013). However, all these additions represent only my opinion and the final decision remains with the author of the review.

Author Response

A first, general remark to all reviewers:

I well agree with the remarks of all Reviewers that the manuscript is  not reflecting the up-to-date situation in terms of the very vast works in the field, having appeared during the last years (since about 2018), nor is it exhaustive in works before that time. In particular, there are only very few applications of LIPSS, compared to the very large range of publications.

However, the intention for writing the present review should be considered:

On the occasion of the Nanomaterials’ Special issue “Dynamics and Processes at Laser-Irradiated Surfaces—A Themed Issue in Honor of the 70th Birthday of Professor Jürgen Reif“, I was asked to contribute an extended review – which, I understood, should be mostly a compilation of OUR contribution to the field, backed by references which I consider as relevant for our work. The second limitation was set by my retirement in 2018, after which I was somehow restricted in following the evolution of the field in full extension. Also, in terms of applications, I limited the work also to fields where we provided actual contributions.

List of changes:

Figs. 1-4,, 7: replaced by figs. with higher resolution

Figs. 5, 11, 22: shifted in position

Fig. 8: improved legend

Fig.10: corrected deformation of axes (X,Y same scale)

Fig 21: the figure had been lost from the manuscript during submission/processing; figure newly introduced into the manuscript

Several incomplete, misleading or bad formulations corrected

Typographical errors corrected

Ref.[98]: Title of publication added

Reviewer 3 Report

The paper provides a review on the topic of the effect of laser irradiation on surfaces. It is well structured and provides an extensive overview of the work that has been done in the field. There are a couple of concerns with respect to the paper.

The first is a large number of self-citations. However, this is perhaps to be expected as the author is the same person to whom the special issue, this paper is written for, is dedicated.

The second is the lack of references for the figures, while I understand that some of these were from the personal archive of the author, if they have been published previously they should be referenced.

Some minor issues:

- Figure 1, the quality of the figure is poor.

- Line 94, Silicon should not be capitalised.

- The sentence starting on line 117 does not make sense

- Line 131, it probably should be ‘results in’ where currently in is missing.

- Line 134, please check heading (. appear to be swapped.

- Line 190, he should be the.

- Line 203, is should be are.

- Line 229, remove ‘the’.

- Line 233, increase should be ‘increased’.

- line 261, I assume THX means thanks?

- Line 281, are these results still unpublished even though the results are 7 years old?

- Line 292, there is an ‘it’ missing at the start of this line.

- There is no Figure 21.

- Line 456, ‘Two’ should be ‘to’.

- Line 468, please check figure number.

- Line 621, what is meant with ‘volume 5’?

Author Response

A first, general remark to all reviewers:

I well agree with the remarks of all Reviewers that the manuscript is  not reflecting the up-to-date situation in terms of the very vast works in the field, having appeared during the last years (since about 2018), nor is it exhaustive in works before that time. In particular, there are only very few applications of LIPSS, compared to the very large range of publications.

However, the intention for writing the present review should be considered:

On the occasion of the Nanomaterials’ Special issue “Dynamics and Processes at Laser-Irradiated Surfaces—A Themed Issue in Honor of the 70th Birthday of Professor Jürgen Reif“, I was asked to contribute an extended review – which, I understood, should be mostly a compilation of OUR contribution to the field, backed by references which I consider as relevant for our work. The second limitation was set by my retirement in 2018, after which I was somehow restricted in following the evolution of the field in full extension. Also, in terms of applications, I limited the work also to fields where we provided actual contributions.

List of changes:

all Figs: References added

Figs. 1-4,, 7: replaced by figs. with higher resolution

Figs. 5, 11, 22: shifted in position

Fig. 8: improved legend

Fig.10: corrected deformation of axes (X,Y same scale)

Fig 21: the figure had been lost from the manuscript during submission/processing; figure newly introduced into the manuscript

Several incomplete, misleading or bad formulations corrected

Typographical errors corrected

Ref.[98]: Title of publication added

Detailed response to Reviewer 3

- Figure 1, the quality of the figure is poor.   see above

- Line 94, Silicon should not be capitalised. done

- The sentence starting on line 117 does not make sense  new sentence

- Line 131, it probably should be ‘results in’ where currently in is missing. corected

- Line 134, please check heading (. appear to be swapped. corrected

- Line 190, he should be the.  corrected

- Line 203, is should be are.  corrected

- Line 229, remove ‘the’.  corrected

- Line 233, increase should be ‘increased’.  corrected

- line 261, I assume THX means thanks? correct, clearer formulation

- Line 281, are these results still unpublished even though the results are 7 years old? In fact, that is the case. The results were presented at the 2016 EMRS meeting, but not otherwise pbulished

- Line 292, there is an ‘it’ missing at the start of this line. corrected

- There is no Figure 21. see above, Figure 21 introduced into the manuscript

- Line 456, ‘Two’ should be ‘to’.  corrected

- Line 468, please check figure number.  corrected

- Line 621, what is meant with ‘volume 5’? see above: "5" should read "V" (corrected)

Round 2

Reviewer 1 Report

The manuscript has been significantly improved. I would suggest it be published.